# Neuronal identity is not static: An input-driven perspective

Nishant Joshi[1], Sven van Der Burg[2], Tansu Celikel[3,4], Fleur Zeldenrust[1]*

1 Donders Institute for Brain, Cognition and Behaviour, Radboud University, Nijmegen, The Netherlands,
2 Netherlands eScience Center, Amsterdam, The Netherlands, 3 Department of Psychology, Georgia Institute of Technology, Atlanta, Georgia, United States of America, 4 Georgia Tech-CNRS IRL, Georgia Institute of Technology - Europe, Metz, France

* fleur.zeldenrust@donders.ru.nl

## Abstract

Neuronal classification based on morphology, electrophysiology, and molecular markers is often considered static. Here, we challenge this view, showing that functional classification depends on input patterns. Using single-cell recordings from layer 2/3 barrel cortex neurons in mice, we compared responses to step-and-hold versus dynamic frozen noise inputs that mimic presynaptic activity. Action potential and waveform-based classifications varied significantly, highlighting the dynamic nature of neuronal identity. To assess the contribution of input versus neuronal attributes toward classification, we analyzed four attribute sets, namely action potential, passive biophysical, adaptation currents, and linear input filters derived via spike-triggered averages (STA). Our findings revealed that the STA, which captures a neuron's selective responsiveness to presynaptic activity, explained the most variance within the population. This highlights input-driven dynamics as key to functional identity, emphasizing the need for physiologically relevant inputs in defining neuronal classes and shifting the focus from static properties to dynamic functional diversity.

## Author summary

Traditionally, scientists have grouped neurons into fixed types based on their shape, electrical activity, or the molecules they express. In this study, we show that this approach misses an important point: a neuron's behavior can change depending on the kind of input it receives. By recording the electrical activity of neurons in the mouse brain and testing them with both a static and a changing input, we found that the way neurons are classified can shift dramatically depending on the input they receive. Our results suggest that to truly understand what makes each neuron functionally unique, we need to look at how they respond to a specific pattern in the input rather than static and firing properties. This means that neuronal identity is not fixed, but is shaped by input, and we encourage future research to focus on these dynamic properties when studying neurons.

**Data availability statement:** All analyzed current clamp and simulation data and the code to analyze and simulate them can be found in this repository: https://doi.org/10.34973/4f3k-1s63. The database is an extended version of the dataset [18,69]. The extracted attributes from the raw data is available at https://doi.org/10.5281/zenodo.17625155, which can be used to replicate the figures and analysis. The code for all figures and analyses is available here: https://github.com/Nishant-codex/single_cell_analysis.git.

**Funding:** This project has received funding from the European Union's Horizon 2020 research and innovation program under the Marie Skłodowska-Curie grant agreement No 860949 (to F.Z and T.C). The funders had no role in study design, data collection and analysis, decision to publish, or preparation of the manuscript.

**Competing interests:** The authors have declared that no competing interests exist.

## Introduction

Neural circuits are composed of diverse neuronal populations that exhibit variability in morphology, molecular composition, and electrophysiological properties. These neurons interact dynamically to process sensory information, support cognition, and drive behavior [1]. A long-standing challenge in neuroscience has been to classify neurons into meaningful functional groups, with traditional approaches relying on intrinsic features such as molecular markers, morphological characteristics, and electrophysiological properties. However, despite significant advances, a consensus on the most informative attributes for neuronal classification remains elusive [2–8]. An often-overlooked dimension in this classification challenge is the role of input dynamics in shaping neuronal function.

Neurons act as spatiotemporal filters, transforming incoming synaptic inputs into output firing patterns. This transformation is governed by an interplay between the structure and dynamics of the input a neuron receives, and the intrinsic membrane processes this input dynamically recruits. Traditional classification approaches, with a focus on static properties, may therefore miss critical aspects of functional diversity that emerge from the interaction between neurons and their presynaptic partners. Recent studies suggest that electrophysiological identity is context-dependent [9–11], varying as a function of the stimulation protocol used to probe neuronal function. However, a direct comparison of neuronal classification under physiologically realistic input conditions remains largely unexplored.

Recent advances in patch sequencing [12–14] enable simultaneous extraction of transcriptomic, morphological, and physiological properties, improving neuronal classification. The Allen Brain Institute dataset [9–11] provides a broad classification of visual and motor cortex neurons using multi-modal features (MET types). However, classification based on electrophysiology remains challenging. Neuronal properties exist on a continuum [3], with unclear boundaries between classes. Moreover, molecularly defined types exhibit overlapping electrophysiological properties [9,11], and intrinsic electrophysiology varies with stimulation protocols. While multi-modal techniques improve classification, they overlook the influence of synaptic input. Neurons receive inputs from thousands of presynaptic neurons shaping their firing properties, necessitating classification under physiologically realistic conditions [15,16].

To address this gap, we investigated how neuronal identity is shaped by the nature of the input a neuron receives. We recorded from layer 2/3 neurons in the barrel cortex and compared their responses under two different stimulation paradigms: a standard step-and-hold (SH) stimulus, which provides a static, artificial input, and a frozen noise (FN) stimulus [17], which simulates natural presynaptic activity. We hypothesized that functional classification of neurons would be stimulus-dependent, with the FN protocol revealing a distinct organization of neuronal diversity that is not captured by the SH protocol.

We analyzed four commonly used sets of electrophysiological attributes— action potential attributes, passive biophysical attributes, adaptation currents, and linear input filters estimated via spike-triggered averaging (STA) to assess their contributions to neuronal classification under dynamic input conditions. Using

multiset-correlation and factor analysis, we found that the linear input filter, which characterizes a neuron's sensitivity to specific input features, was the most informative attribute for understanding neuronal functional variance, i.e, it is the most relevant attribute to be used for clustering neurons. This finding challenges the traditional view that neuronal identity is a static property, emphasizing the importance of input dynamics on functional diversity.

By demonstrating that neuronal classification is highly dependent on input dynamics, our study highlights the need to incorporate physiologically relevant stimuli when defining neuronal types. Our findings suggest that neurons should not be categorized based solely on static features but rather on how they process and respond to dynamic synaptic input. This perspective has broad implications for both experimental and computational neuroscience, urging a paradigm shift toward input-dependent models of neuronal function.

## Materials and methods

### Ethics statement

The data used in this research was previously published and made freely available to the community [18] and [19]. All the experimental work, as outlined in the cited articles, were carried out in compliance with the European directive 2010/63/EU, the national regulations of the Netherlands, and international standards for animal care and use. The experiments were approved by the Institutional Animal Care and Use Committee (IACUC) at Radboud University (AVD10300201850485).

### Slice electrophysiology

Data acquisition procedures, the details of the in vitro slice preparation, intracellular access to anatomically targeted neurons, data digitization, and preprocessing have been described in detail elsewhere [18,20–23]. In short, Pvalbtm1(cre)Arbr (RRID:MGI:5315557) or Ssttm2.1(cre)Zjh/J (RRID:IMSR_JAX: 013044) mice, including both females and males, were obtained from local breeding colonies and studied after the maturation of evoked neurotransmitter release in the primary somatosensory cortex [24].

Mice were anesthetized with Isoflurane (1.5 mL/mouse) before extracting tissue, and coronal slices of the primary somatosensory cortex (barrel subfield) were prepared. The brain was removed, and 300 μm-thick coronal slices were made. Slices were then incubated in artificial cerebrospinal fluid (aCSF) (120 mM NaCl, 3.5 KCl, 10 glucose, etc.), aerated with 95% $O_2$/5% $CO_2$ at 37°C, and then at room temperature after 30 minutes.

Whole-cell electrophysiological recordings were performed with continuously oxygenated aCSF. The barrel cortex was localized, and cells in the supragranular layers were patched under 40x magnification using HEKA EPC 9 and EPC10 amplifiers with Patch Master software. Patch-clamp electrodes were pulled from glass capillaries (1.00 mm external diameter, 0.50 mm internal diameter) and used with 5-10 MOhm resistance, filled with intracellular solution (130 mM K-Gluconate, 5 $KCl$, 1.5 $MgCl_2$, etc., pH adjusted to 7.22 with $KOH$). Data were band-pass filtered at 0.1-3000 Hz before storage for offline analysis.

**Step and hold (SH) protocol.** The Step and Hold protocol was set up in a current clamp configuration, where the resting membrane potential was set to -70 mV before current injection into the soma of the neuron. In total, 10 current step injections, each 500 ms long, were performed. The current steps were fixed and were depolarizing in nature, ranged from 40 pA to 400 pA with an inter-sweep interval of 6.5s. The stimulus was repeated 1 to 3 times for neurons. The drifts encountered were not corrected for.

**Frozen Noise (FN) protocol.** The Frozen Noise protocol involves injecting a somatic current generated by a simulated Poisson neural network of 1000 neurons. Each neuron fires Poisson spike trains modulated by a binary hidden state, which represents the presence or absence of a stimulus and switches according to a Markov process. The hidden states have a time constant of either 250 ms or 50 ms. The input generated with a time constant 250 ms is used for neurons that don't respond to a rapidly fluctuating input, these are suspected to be excitatory in nature. The spike trains are

convolved with an exponential kernel (decay time 5 ms) and linearly combined with weights $w_i = log(q_{on}^i / q_{off}^i)$ where $q_{on}^i$ and $q_{off}^i$ are the neuron's firing rates in the on and off states, respectively. The resulting current is scaled and injected into the neuron, and the membrane potential is recorded at 20 kHz for 360 seconds. During the FN recording process, neurons were initally observed under a microscope and then were probed with an initial step input. Based on the response, the FN recordings were performed either with 250 ms FN input when neurons showed a slow firing response (<50 Hz) to the probe input along with a triangular shaped soma, and with a 50 ms input otherwise [19]. The protocol is repeated for control (aCSF) and drug (agonist) conditions, and the data are saved as MATLAB structs with metadata including cell type, baseline, scaling factor, and experimental parameters. See [17,18] for further details.

## Method details

**Feature extraction.** Feature extraction is performed separately for SH and FN protocols to classify neurons using different inputs and to determine cell class with a realistic stimulus. For the first part, which is comparing classification across protocols, we collect waveforms and Action potential attributes for comparing SH and FN protocols using a subset of the data (186 neurons). As not all cells were recorded with both types of input protocols, this subset was chosen to match the cell IDs across the two protocols. For the second part, which compares different physiological attributes for the FN protocol, we collect waveforms, action potential features, biophysical features, and the STA for each control trial for each cell in the FN dataset. We discard trials that contain recording artifacts (observed distortions or high-frequency noise), in total 11 neurons. Since there were multiple drug and control trials available for each neuron, we always took the first control trial for this study as this prevented any residual effect from the drug condition after washing. The total number of neurons used for this part of the study was 312.

**Spike waveforms.** Spike waveforms were extracted for each neuron from both SH and FN protocol trials (this could not be done for all the neurons as some neuron IDs could not be matched between SH and FN protocols due to missing metadata). Firstly, we identified the spike times in each trial in each protocol. This was done by identifying peaks in the membrane potential trace using the Scipy Findpeaks function [25] with *height* = $20mV$, and *distance* = $80ms$ as the chosen hyperparameters. Next, we cut the spike waveform by defining a window of 2 ms before and 3 ms after each spike peak to get a 5 ms long waveform for each spike. We ignored the spikes that had an ISI lower than 3 ms as we were interested in non-bursting type spike shapes. We then average over all the waveforms for each trial to get an average waveform shape in both FN and SH protocols respectively.

Extracted Waveforms for the second part were 10 ms long (5 ms before and after the spike peak), to incorporate the subthreshold dynamic before reaching the threshold of the neuron in the waveform classification as well. We observed some variability in the baseline membrane potential values as well as the slope of membrane potential before the threshold value was reached.

**Action potential features.** Action potential features were extracted to study the variability in spike-related dynamics across individual neurons. These features were designed based on their suitability for comparing neurons in FN and SH protocols. These Action potential features were divided into three categories: spiking dynamics, spike threshold, action potential height and width.

**Spiking dynamics.** This includes the following features:

- **Current at the first spike**: We take the current amplitude when the membrane potential crosses the threshold for the first time in the trial for FN protocol. For SH protocol, it is the current step that produces the spike for the first time.
- **AP count**: We count the number of spikes over the entire trial length of 360 seconds for the FN trial and 500 milliseconds for the SH trial which is the duration for the current onset.
- **Time to first spike**: We measure the time (in milliseconds) it takes for the neuron to fire the first action potential. For the SH case, we take the lowest amplitude step where a spike is observed.

- **Firing rate**: We calculate the firing rate as the number of spikes per second. For the FN case, we take the entire length of 360 seconds of the trial and for the SH case, we take the duration of the current onset which was 500 milliseconds for the highest current step (400 pA).

$$fr = \frac{N_{spikes}}{T} \tag{1}$$

where $N_{spikes}$ is the total number of spikes in the trial and T is the total length of the trial.
- **Interspike Interval**: Interspike interval is the duration between two spikes $t_{spike_{n+1}} - t_{spike_n}$, where $t_{spike_{t+1}}$ and $t_{spike_t}$ are spike times for spikes n+1 and n. We take the interspike interval for all the spikes in the trial for both FN and SH protocols and calculate the mean, median, minimum, and maximum values.
- **Instantaneous rate**: Instantaneous rate is defined as the reciprocal of the average of the Interspike interval.

$$inst.firing\_rate = \frac{1}{t_{spike_{n+1}} - t_{spike_n}} \tag{2}$$

### Spike threshold

- We measure the threshold of each spike from the trial as described in [26]. The threshold is defined as the voltage V at the spike onset when the first derivative of the membrane potential $dV/dt$ reaches 25 mV/ms for the first time. We take the first threshold value of the trial as well as the mean, median, maximum, and minimum values of the thresholds from the entire trial length.

### Action potential height and width

- **Width**: Spike width is calculated as the time it takes between when the membrane potential reaches the AP threshold and when the membrane voltage goes below the threshold after the spike peak. We calculate the width for each spike in the trial for both FN and SH protocols. We calculate the mean, median, maximum, and minimum for all the values obtained from a trial.
- **Amplitude**: Spike amplitude is calculated as the difference between AP peak and AP threshold value. We calculate the amplitude for each spike in the trial for both FN and SH protocols. We calculate the mean, median, maximum, and minimum for all the values obtained from a trial.

**Biophysical feature extraction using GLIF model.** Since it is not possible to empirically observe the passive biophysical properties of the cell just using the membrane potential, we fit a GLIF model to the recordings, which can capture universal spiking and sub-threshold dynamics [27]. The following equations define the GLIF model:

$$C\dot{V}(t) = -g_L(V(t) - E_L) - \sum_{\hat{t}_j < t} \eta(t - \hat{t}_j) + I(t), \tag{3}$$

where $V(t)$ is the membrane potential, C is the membrane capacitance, $gL$ is the leak conductance, $EL$ is the resting potential, and $\eta(t)$ is the adaptation current triggered by a spike event. Spikes are stochastically produced by a point process that represents conditional firing intensity $\lambda(t|V, V_t)$ that is dependent on the instantaneous difference between the membrane potential and voltage threshold given by:

$$\lambda(t|V, V_t) = \lambda_0 * exp\left(\frac{V(t) - V_T(t)}{\Delta V}\right), \tag{4}$$

where $\lambda_0$ is the base firing rate in Hz, V(t) is the membrane potential, and $V_T(t)$ is the moving spike threshold and $\Delta V$ controls the sharpness of the exponential threshold function. The probability of a spike $\hat{t}$ between a time interval $t$ and $\Delta t$ is given by the following equation (based on [28]):

$$P(\hat{t} \in [t, t + \Delta t]) = 1 - exp(- \int_t^{t+\Delta t} \lambda(s)ds) \approx 1 - exp\left(- \lambda(t)\Delta t\right), \tag{5}$$

The dynamics of the firing threshold $V_T(t)$ are given by:

$$V_T(t) = V_T^* + \sum_{\hat{t}_j < t} \gamma(t - \hat{t}_j), \tag{6}$$

where $\gamma$ is the stereotypical movement of the spike threshold after a spike and $V_T^*$ is the threshold baseline.

The method for fitting this neuron model to a membrane potential recording is divided into the following steps:

**Preparation step:** A 100-second window from the initial part of the trial is taken as the training set for the fitting, [27] shows that a longer trial length doesn't improve the fit. Spike times and waveforms are also extracted for the preparation step of the fitting procedure.

**Step 1. Fitting the reset voltage:** The waveforms extracted in the preparation step were averaged, and then the Reset voltage $V_{reset}$ was extracted using the averaged waveform by setting an arbitrary refractory period $t_{ref}$ and taking the membrane potential value at $t_{\hat{i}} + t_{ref}$, where $t_{\hat{i}}$ is the spike peak. The refractory period is chosen to be always lower than the minimum inter-spike interval, we chose the refractory period of $t_{ref} = 4ms$ in this case.

**Step 2. Fitting sub-threshold dynamics:** The voltage dynamics in Eq (3) are given by parameter set $\theta_{sub}$ ={C, $g_L$, $\eta$, and $E_L$}, by fitting the temporal derivative of the data $\dot{V}_{data}$ in the model, we can extract the set of passive parameter set $\theta_{sub}$ for the data. Firstly, we can write the adaptation current $\eta$ as a linear sum of rectangular basis functions [29].

$$\eta(t) = \sum_{k=1}^{K} a_k f^{(k)}(t), \tag{7}$$

Using the fact that the voltage dynamics are approximately linear in the subthreshold regime, $\theta_{sub}$ parameter set can be extracted using a multi-linear regression between $\dot{V}_{data}$ and $\dot{V}_{model}$. For this, we created a training set $V_{data}^{sub}$ where we removed the spike waveforms from $V_{data}$, $V_{data}^{sub} = \{V_{data}(t)|t \notin (t_{\hat{i}} - 5ms, t_{\hat{i}} + t_{ref})\}$, where $t_{\hat{i}}$ is the spike times. The regression problem can be stated as:

$$\theta_{sub} = (X^T X)^{-1} X^T \dot{V}_{data}^{sub}, \tag{8}$$

where $X^T$ is a matrix representing parameter values at different time points, its row elements $x_t^T$ are of the following form

$$x_t^T = [V_{data}^{sub}, 1, f^{(1)}(t), f^{(2)}(t), ..., f^{(K)}(t)], \tag{9}$$

**Step 3. Fitting the spike probability:** For fitting the spiking probability to the data, we need to extract parameters defining the dynamics of the threshold Eq (6). The stereotypical shape of the adaptation current threshold movement can be expanded as a sum of rectangular basis functions as follows [29]:

$$\gamma(t) = \sum_{p=1}^{P} \gamma_p f^{(p)}(t), \tag{10}$$

We use the parameters obtained in the previous steps to compute the subthreshold membrane potential of the model using numerical integration of Eq (3). We set $\lambda_0 = 1$ Hz and all the threshold parameters $\theta_{thr} = \{\Delta V, V_T^*$, and $\gamma(t)\}$ are extracted by maximizing the likelihood function of the following form based on the experimental spike train:

$$\hat{\theta}_{thr} = \underset{\theta_{thr}}{\arg\max}\left[\sum_{t \in \{\hat{t}_j\}} y_t^T \theta_{thr} - \Delta T \sum_{t \in \Omega} exp(y_t^T \theta_{thr})\right], \tag{11}$$

Where $\Omega = \{t | t \notin (\hat{t}_j, \hat{t}_j + t_{ref})\}$,

The subthreshold fit is examined by comparing the variance explained $R^2$ of the subthreshold membrane potential trace $V$ between the data and the model. All the models chosen for clustering had an $R^2$ value > 0.7. The sets $\theta_{clustering} = \{g_L, \Delta V, C, V_T^*, E_L, V_{reset}\}$ are the parameters that are extracted from the model that is used in the clustering procedures.

**Spike triggered average.** The STA is the average shape of the stimulus that precedes each spike. We extracted the STA using the following equation given by [30]:

$$STA = \frac{1}{N}\Sigma_{n=1}^N \vec{s}(t_n), \tag{12}$$

where $t_n$ is the $n^{th}$ spike time, s is the stimulus vector preceding the spike for a fixed time window of 100 ms, and N is the total number of spikes. Before clustering, we standardize (i.e. z score) and then normalize the STA vector with an $L_2$ norm. We didn't use any kind of whitening or regularization to calculate the STA.

## UMAP + Louvain clustering

Conventional clustering algorithms such as K-means do not perform well in high dimensional spaces (p»N, where p is the dimension of data and N is the number of samples) due to the curse of dimensionality [31], and therefore need a pre-processing dimensionality reduction step. Addressing this issue, a non-linear dimensionality reduction algorithm such as UMAP [32] creates a high-dimensional graph representation of the data which can be utilized for clustering using a graph-based clustering method such as Louvain clustering [33] or ensemble clustering [34]. This method utilizes the high dimensional space of the data for clustering. As shown by [35] using the WaveMAP algorithm, the UMAP+Louvain community detection algorithm has been successful in finding neuron types based on extracellular recordings.

**UMAP.** Universal Manifold Approximator (UMAP) is a non-linear dimensionality reduction technique that preserves local and global relationships between data in high dimensional space [32]. It is divided into two steps, the first step is creating a k-nearest neighbor graph and the second step is to generate a low-dimensional representation that is similar to the high-dimensional graph structure.

We used the Scikit-learn UMAP-learn software package [36] to extract the embedding and the graphs.

**Louvain community detection.** The Louvain community detection algorithm [33] maximizes modularity among the identified groups in a graph. Modularity can be defined by the following equation:

$$Q = \frac{1}{w}\sum_{i,j}\left[A_{i,j} - \gamma\frac{d_i^+, d_i^-}{w}\right]\delta(c_i, c_j) \tag{13}$$

where $A_{i,j}$ is the adjacency matrix, $k_i = \sum_j A_{ij}$ is the sum of the weights of the edges attached to the vertex i, $c_i$ is the community for vertex i, $d_i$ is the degree of node i, $d_i^+$ and $d_i^-$ are the in degree and out degree for node i,$\delta(u, v)$, Kronecker symbol, is 1 if u=v and 0 otherwise, $w = \sum_{i,j} A_{i,j}$ and $\gamma > 0$ is the resolution parameter. For Louvain graph-based clustering we used the implementation from the Scikit-network software package [37].

The clustering approach can be summarized in the following steps:

1. The high dimensional k-neighbor graph is obtained by the first step of the UMAP algorithm using data vectors that are first standardized and then normalized using the $L_2$ norm. The nearest neighbor and distance parameters for UMAP were 20 and 0.1 respectively. This is to ensure a compact embedding and a clear clustering.
2. Using the graph obtained in the first step, we perform Lovain community-based clustering, using the resolution parameter $\gamma$ that maximizes the modularity score, and the corresponding community/partition is chosen as the final cluster labels.
3. Using the cluster labels found in the second step, we color the individual points in distinct colors on the low-dimensional UMAP representation.

This unsupervised clustering approach was effective in capturing the global structure of the high dimensional space across attributes such as waveforms, adaptation current, and Spike Triggered average. The clusters found using this method were robust even when clustering was repeated with a sub-sample of the data and while iteratively removing the features.

**Cluster stability and parameter selection.** Cluster stability is tested by clustering a 90% sub-sample of the data chosen at random and repeating the procedure 25 times for each resolution parameter, varying from 0 to 5 with a step of 0.5. The modularity score is calculated for each resolution parameter and finally, the resolution parameter is chosen for which the modularity score is maximal. Variation in the number of clusters is observed for each resolution parameter and is contrasted with the modularity score. Clustering robustness is also tested by repeating the procedure above while excluding one feature at a time for Action potential feature clustering and passive biophysical clustering.

**Cluster likelihood comparison.** Cluster likelihood between two sets of labels was calculated in two steps, firstly we created a contingency matrix such that $C_{i,j}$ contained the number of times neurons classified in cluster $i$ in the SH protocol classified as cluster $j$ in the FN protocol, such that each row contains the division of elements of cluster $i$ in the SH protocol into all the clusters of the FN protocol. Secondly, to get the likelihood, we divided each row by the total count of neurons in cluster $i$ in the SH protocol.

$$P(j|i) = \frac{C_{i,j}}{\sum_j C_{i,j}} \tag{14}$$

Here $P(j|i)$ is the probability of the neuron classifying into FN cluster $j$ given it is classified in SH cluster $i$.

**Cluster similarity measures.** We measured the similarity between two given cluster assignments using the Adjusted Random Index (ARI) and Adjusted Mutual Information (AMI) Score. Both of these measures amount to change agreements between two clusters. The ARI (range 1-1) measures the pairwise relationships between clusters and the AMI (range 0-1) measures the overall information shared between the two clusters.

We used the scikit-learn Python package [38] to calculate the ARI and AMI measures.

**Ensemble Clustering for Graphs (ECG).** We used Ensemble Clustering for Graph algorithm [34] to validate the clusters found using the Louvain Community detection algorithm. It is a two-step algorithm, the first step called the generation step consists of producing a $k$–level 1 partition $P = \{P_1, P_2, ...P_k\}$ by running the first pass of the Louvain clustering algorithm with random vertices on the initial graph G= (V, E). The second step, also known as the integration step, consists of running the Louvain algorithm on a weighted version of the initial graph. Where the weights are the weight of an edge given by

$$W_p(u, v) = \begin{cases} w_* + (1 - w_*) * \left( \frac{\sum_{i=1}^{k} v_{p_i}(u,v)}{k} \right) & , \text{if } (u,v) \text{ is in 2-core of G} \\ w_* & , \text{otherwise} \end{cases} \tag{15}$$

where $0 < w_* < 1$ is the minimum ECG weight and $v_{p_i}(u, v) = \sum_{j=1}^{l_i} 1_{C_i^j}(u) \cdot 1_{C_i^j(v)}$ shows if the vertices u and v co-cluster in the same cluster of $P_i$ or not. Thus it takes advantage of multiple instances of the Louvain clustering algorithm to make a clustering based on consensus. We used the implementation provided by [35] for comparing the waveform clustering for SH and FN protocols. The graph used for Ensemble clustering was the same as in the original clustering using the UMAP algorithm.

## Quantification and statistical analysis

**MANOVA.** We measured the significance between excitatory and inhibitory action potential and passive biophysical feature vectors using a one-sided MANOVA. We present the following statistics: Wilk's lambda, Pillai's trace, Hotelling-Lawley trace, and Roy's greatest root using the Stats Model python package [39].

**Canonical correlation analysis.** To perform a post-hoc analysis on action potential and passive biophysical attributes, to find the importance of each feature, we used a canonical correlation analysis. Which is a method to find a linear combination of features between two datasets that maximizes the correlation between them. It is a deterministic method that results in canonical variates of the two datasets that are maximally correlated. Since the excitatory/inhibitory populations were different in numbers, we repeated the SHA procedure 10 times with random sampling from the larger group to make the population size equal between excitatory and inhibitory groups. We then obtain loadings for each dataset by averaging over the 10 repetitions, which represents the correlation of a feature with the canonical variate. We used the Scikit learn python package [38] for performing the SHA procedure.

**Welch's ANOVA.** To compare the significance of cosine similarity measures for excitatory and inhibitory populations, we first calculated the cosine similarity matrix comparing the excitatory and inhibitory populations separately and a third matrix comparing excitatory with inhibitory populations. We take the upper triangular part of the excitatory and inhibitory within-population comparison matrix and the entire excitatory vs inhibitory matrix for the significance test. The Welch's ANOVA test was used to compare the three groups of cosine similarity for each attribute with one categorical variable namely, excitatory or inhibitory. This test was chosen because the populations for each attribute were heterodrastic and of unequal size. We performed a post-hoc Games-Howell test to determine which group was significantly different, this was done because the variances across groups were heterodrastic. The significance levels are reported in the figures based on the p-values obtained by the post-hoc test. We used the Stats model python package [39] for calculating Welch's ANOVA.

## Multi-set correlation and factor analysis

The Multi-set Correlation and Factor Analysis (MCFA) was performed using the procedure as described by [40], and the accompanying software was used for the analysis [41]. The problem can be stated as follows, let $\{Y^m\}_{m=1}^M$ be a set of M attributes extracted from the electrophysiological data, each with dimension $NxP_m$, where N is the number of samples and $P_m$ is the dimension of each attribute (Action potential parameters ($P_1$ =22), passive biophysical parameters ($P_2$ =6), Spike triggered Average ($P_3$ =2000), and Spike triggered current ($\eta$) ($P_4$ =10000)). Each attribute set can be modeled as having a contribution from two factors, a shared and a private factor respectively, as shown below.

$$z_n \sim N(0, I_d) \tag{16}$$

$$x_n^m \sim N(0, I_{k_m}) \tag{17}$$

$$y_n^m \sim N(W_m z_n + L_m x_n^m, I_{k_m}, \Psi_m) \tag{18}$$

Here $z_n$ is the shared factor of dimension $d$, and $W_m$ is the shared space loading matrix of shape $P_m \times d$. $k_m$ is the dimension of each private mode. $x^m$ is the private space for each attribute m of dimension $k_m$, $L_m$ is the private space loading of shape $P_m \times k_m$. $\Psi_m = diag(\psi_m^1, ..., \psi_m^{\rho_m})$ are the diagonal residual covariance matrices. Given $Y$, $d$, and $k_m$ the goal is to find hidden factors $z_n$, $x_n^m$, and loading matrices $W_m$, $L_m$. This is achieved by the Expectation Minimization (EM) method [40].

We center and scale all variables as in [40] and initialize the loading matrices similarly to the original method, using the pairwise correlations with average variance constraint initialization. To model the shared latent space $z_n$, we chose the most informative PCA components based on Marchenko Pasteur Law to control over-fitting [40], which states for any normalized dataset ($\mu = 0$, $\sigma = 1$), the principal components with eigenvalues above $\lambda_m = 1 + \sqrt{p_m/N}$ are considered non-noise. We set the size of the private space $k_m = 1$ for passive biophysical parameters due to its relatively lower dimension and $k_m = 2$ for the other attributes. After initializing all the variables, we run the expectation minimization (EM) algorithm to obtain $z_n$, $x_n^m$, $W_m$, and $L_m$ matrices.

For feature j of mode m, the variance explained by a shared feature d is given by $W_m^{(j,d)2}$. Similarly, the variance explained by the $k^{th}$ private factor of feature j of mode m is given by $L_m^{(j,k_m)2}$. The total variance explained for a mode by a shared factor is given by $\sum_j W_m^{(j,d)2}$, similarly, the total variance explained by the private factor $k_m$ is $\sum_j L_m^{(j,k_m)2}$. Hence, the total variance explained by all shared and all private factors is given by $\sum_{j,d} W_m^{(j,d)2}$ and $\sum_{j,k_m} L_m^{(j,k_m)2}$.

The relative feature importance is given by the cross-correlation of columns of the posterior mean of Z on observing a single mode denoted by $\hat{Z}_m$ [40]

$$\hat{Z}_m = E[Z|W_m, \Psi_m, L_m, Y_m] = Y_m(W_m W_m^T + L_m L_m^T \Psi_m)^{-1} W_m \quad (19)$$

$$S_d = cor(\hat{Z}_1^{(:,d)}, ..., \hat{Z}_m^{(:,d)}) \quad (20)$$

## Results

We aimed to observe the contribution of the input type in explaining functional classification and to explore what a physiologically realistic stimulus reveals about the functional classification of a neuron. For the first aim, we researched the stimulus dependence of neural classification using two different sets of classification features: 1) action potential waveforms and 2) other action potential attributes, such as the spike threshold and spiking dynamics. Next, to research the second aim, we investigated which attributes are the most informative about neuronal heterogeneity under a physiologically realistic stimulation protocol. Therefore, we performed classifications based on four different attribute sets that capture 1) action potential (AP) attributes, 2) Passive biophysical (PB) attributes, 3) Adaptation current (AC), and 4) linear input filter through a Spike Triggered Average (STA). Finally, we used a method known as Multi-set correlation and Factor Analysis (MCFA) to compare the variance explained by the shared structure across these four attribute sets.

### Stimulus dependence of neural classification

We first aim to understand the role of a neuron's input in the functional classification of neural populations. For this, we analyze single-cell patch-clamp recordings [18,19] recorded with two different input conditions: 1.) a Step and Hold and 2.) a Frozen Noise protocol (see Methods). We want to understand the influence of a physiologically realistic FN input on classification using commonly used features in contrast to SH input. To contrast the heterogeneity between SH and FN input conditions, we compare classification in these two input conditions using the neuronal spike waveforms and action potential attribute sets and also measure the similarity in the waveforms and action potential attributes between each cluster across the two input protocols.

**Stimulus dependence of neural classification using intracellular action potential waveforms and attributes.** To understand whether waveform as well as action potential attribute based classification differs under FN and SH input conditions, we analyzed control (aCSF) trials from a total of 186 *in-vitro* whole-cell patch-clamp recordings, where the same cell was recorded under two different input protocols: SH and FN. It has been shown that excitability measures such as total spike count and AMPA conductance threshold (dynamic) versus rheobase (static) have a low correlation between static and dynamic stimulus conditions [42,43]. This motivated us to examine whether the classification of neurons into functional types depends on the stimulation protocol used. For waveform-based classification, we extracted waveforms of equal length (5 ms) from both FN and SH trials, standardized them (subtracting the mean and scaling to unit variance), and then normalized the data using an L2 norm. Similarly, for action potential attribute-based classification, we extracted attributes that incorporate, among others, spiking dynamics, spike threshold, and action potential height and width; a total of 22 attributes (see Methods). This attribute set was designed to enable comparison across input protocols. We used descriptive statistics such as mean, median, minimum, and maximum values for some of the features to capture their distribution within each trial. The data were then standardized and normalized as done with the waveforms.

Next, we apply an unsupervised high dimensional clustering algorithm that combines UMAP and Louvain Community detection (see Methods) and found 7 clusters for SH trials as well as 7 clusters for FN trials (Fig 1A and 1B) respectively. A sample trace from FN and SH trials are shown in Fig 1C and 1D. Applying the same clustering method on the 22 action potential attribute set, we found 7 clusters for SH and FN trials respectively (Fig 2A and 2B). The AP attributes for each trial (thin line) along with the mean for each class (thick line) is shown Fig 2D and 2E. On visual inspection, we observe that the means for each cluster (thick line) in both SH and FN protocols are non-overlapping for all the 3 attribute sets (spiking dynamics, spike threshold, and AP height and width) respectively. Suggesting that each cluster has distinct action potential attributes. We also measured the stability of the clusters (see Methods) against changing the hyperparameter (resolution parameter) for the unsupervised method. We found the clusters to be stable (low standard deviation) for the chosen resolution parameter (chosen at a value of 1.0) (See Fig 1G and 2C). A 5 second long sample trace of each FN waveform class along with it's corresponding SH trial is shown in S7 Fig.

We compared waveform shapes as well as action potential attribute sets across clusters by creating pairwise cosine similarity matrices between FN and SH protocols and then taking the average of the sub-matrix for each SH and FN cluster pair. The average similarity across waveform clusters is summarized using a heatmap (Fig 1E and 1F), and a similar heatmap is drawn for action potential attributes (Fig 2F). We first performed this comparison for each protocol separately and found that some cluster waveform shapes were similar (cosine similarity >0.9) to their immediate neighbors (e.g., clusters 7, 3, and 4 in SH and clusters 6, 3, and 4 in FN; Fig 1E). Comparing waveform shapes between SH and FN, we found that narrow-width clusters for the SH protocol (4, 3, and 7) were highly similar to narrow-width clusters for the FN protocol (6, 2, and 3). For the action potential attribute sets, none of the SH clusters showed a high similarity (>0.9) with FN clusters, suggesting that action potential attributes differ substantially as a result of the input protocol (Fig 2F).

We projected all the SH and FN waveforms together onto a single embedding space and observed that they were distinct (S1A Fig), suggesting that waveform shapes are not fixed across protocols. We then tested for potential drift in the recordings by dividing each 360-second trial into two parts and comparing the waveforms between the first and second halves by projecting them onto the same 2D UMAP space. The two halves overlapped, suggesting that waveform shapes remained consistent throughout the trial. We also overlaid the UMAP embedding for each action potential attribute set for the SH and FN protocols (S1C Fig) and found that the SH and FN feature manifolds were entirely different.

Finally, we calculated the likelihood that neurons in one of the SH waveform-based clusters would cluster together in one of the FN clusters (see Methods and Fig 1H). For each SH waveform cluster, the likelihood of clustering in one of the FN clusters was broadly distributed without a strong majority, suggesting that neurons grouping together under the SH protocol do not group similarly under the FN protocol. We further quantified the clustering agreement between the SH and FN clusters using the adjusted Rand index (ARI) and adjusted mutual information (AMI). Both measures

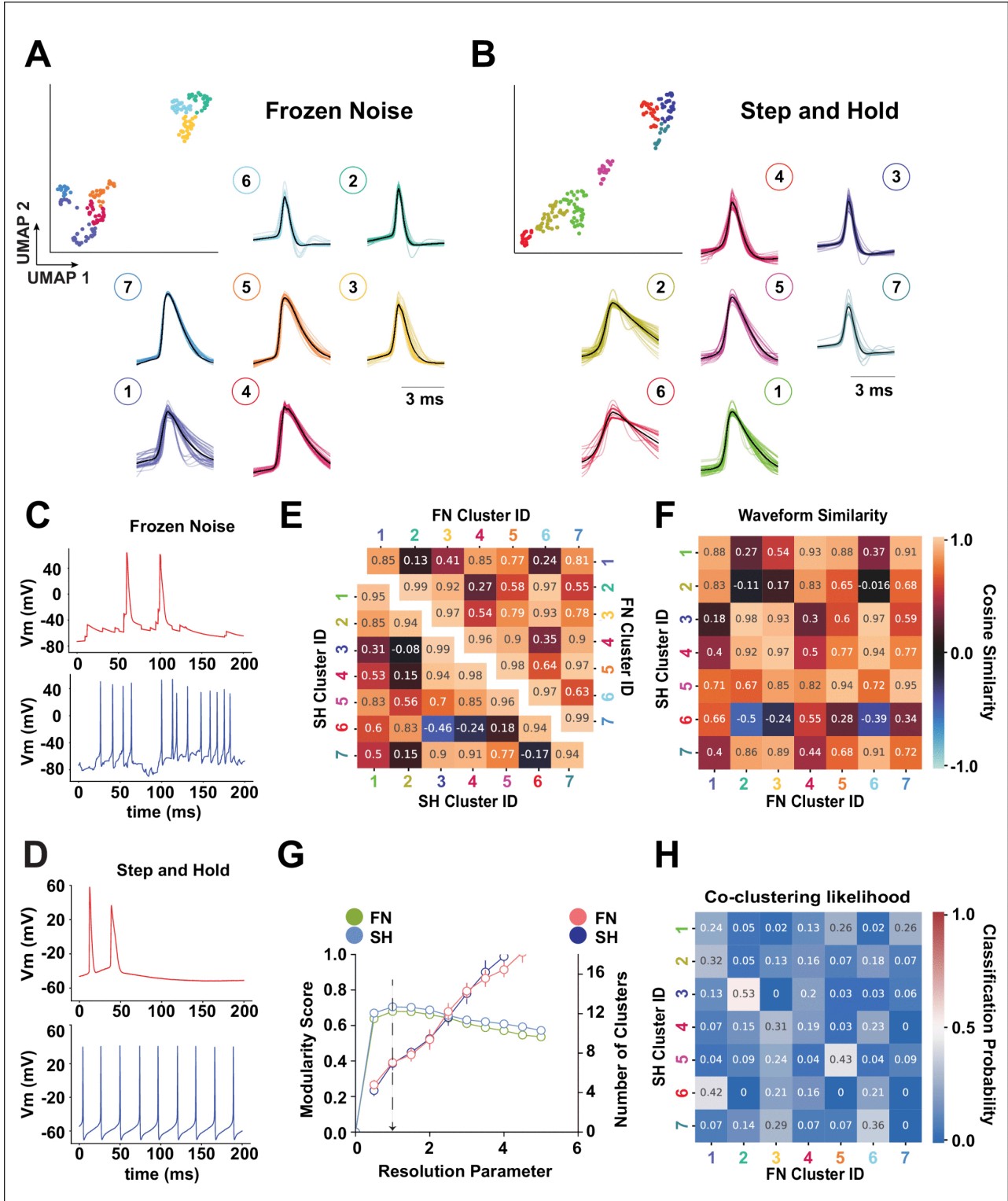

**Fig 1**. **Intracellular waveform-based neural identity depends on the stimulus type.** Each of 186 cortical neurons was recorded under two stimulation protocols: Step-and-Hold (SH) and Frozen Noise (FN). **(A-B)** UMAP projections of spike waveforms from FN and SH protocols, colored by cluster identity (top) with their corresponding average waveform shapes (bottom). **(C-D)** Representative 200 ms traces from FN and SH recordings showing

broad- (red) and narrow-width (blue) spike waveforms. **(E)** Within-protocol cosine similarity matrix showing inter-class waveform similarity for SH and FN clusters. **(F)** Cross-protocol cosine similarity matrix comparing waveform shapes between SH and FN clusters. **(G)** Cluster stability analysis showing mean $\pm$ SD modularity (left axis) and number of clusters (right axis) across resolution parameters; the black arrow marks the chosen value (1.0). **(H)** Likelihood of SH class membership mapping to FN classes. Low ARI (0.085) and AMI (0.133) scores indicate inconsistent class correspondence between protocols.

were low ($ARI = 0.085$, $AMI = 0.133$), confirming that neurons based on their waveform cluster differently between SH and FN protocols. For action potential attribute sets, we found a similar pattern: the likelihood of SH cluster membership corresponding to a specific FN cluster was broadly distributed. The corresponding scores were also low ($ARI = 0.149$, $AMI = 0.206$). To verify that the found clusters were not biased by the method used, we repeated the waveform clustering analysis using the ensemble clustering method (S2 Fig). Taking the average of the highest values for each row in the co-clustering matrix, we found 84% correspondence for FN waveforms and 96% for SH waveforms between the wave map and ensemble clustering. Comparing the ensemble clustering with Louvain clustering using the adjusted mutual information score ($AMI^{FN}_{Louvain \, vs \, Ensemble} = 0.765$, $AMI^{SH}_{Louvain \, vs \, Ensemble} = 0.736$), we found a high level of similarity for both SH and FN protocols.

In conclusion, clustering neurons into cell classes based on their waveforms results in different classifications under SH and FN input protocols, reflecting differences in waveform shapes due to the stimulation protocol. This shows that waveform-based neuronal identity is stimulus-dependent. Similarly, neurons cluster differently between SH and FN protocols based on action potential attributes, suggesting that action potential attribute based clustering of neuronal populations is also input-dependent.

## Functional classification of neurons stimulated by the FN protocol based on different feature sets

We have shown in the previous section that neuronal classification based on waveforms and action potential attributes is input-dependent. We want to expand our understanding of which features result in distinctive functional classifications within the FN-stimulated neurons. To understand the variance in the neuronal population captured by different attribute sets, we perform classifications based on four different attribute sets that capture 1.) the commonly used action potential attributes, 2.) passive biophysical attributes, 3.) adaptation attributes, and 4.) linear input filters approximated using STA, to assess input feature selectivity.

Ample experimental evidence suggests that cortical neurons can be divided into two broad functional categories [44], namely excitatory (glutamatergic) and inhibitory (GABAergic), based on the type of effect (either excitation or inhibition) they have on their post-synaptic neurons [8]. Excitatory and inhibitory neurons have also been found to have distinct electrophysiological properties and thus are known to perform different functions. We therefore subdivide our data into excitatory and inhibitory groups to study the diversity within and across populations. Previous studies have associated neuronal waveform shapes with functional identity [35,45]. The broad and narrow spike-width neurons have also been found to have a characteristic firing statistic, i.e. neurons with narrow-width waveforms were found to have a high firing rate (putatively inhibitory), and neurons with broad-width waveforms were found to have lower firing rates (putatively excitatory) [15,46,47] in the barrel cortex. This suggests that inhibitory neurons can be putatively characterized by narrow width and high firing rate and that excitatory neurons can be putatively characterized by broad width and low firing rate. Also, barrel cortical excitatory neurons are more adaptive compared to inhibitory neurons [48]. Based on this reasoning, we partitioned our data into a putative excitatory and an inhibitory population.

We extract the average intracellular waveforms from the entire dataset, 312 cells in total. Note that the number of neurons under analysis is much larger than in the previous sections because only a small subset of FN trials had a matching SH recording. We then apply the UMAP+Louvain algorithm to classify intracellular waveforms and find 8 clusters. (Fig 3A) shows the UMAP projection of all waveforms with their corresponding cluster label colors. Next, we plot the distribution

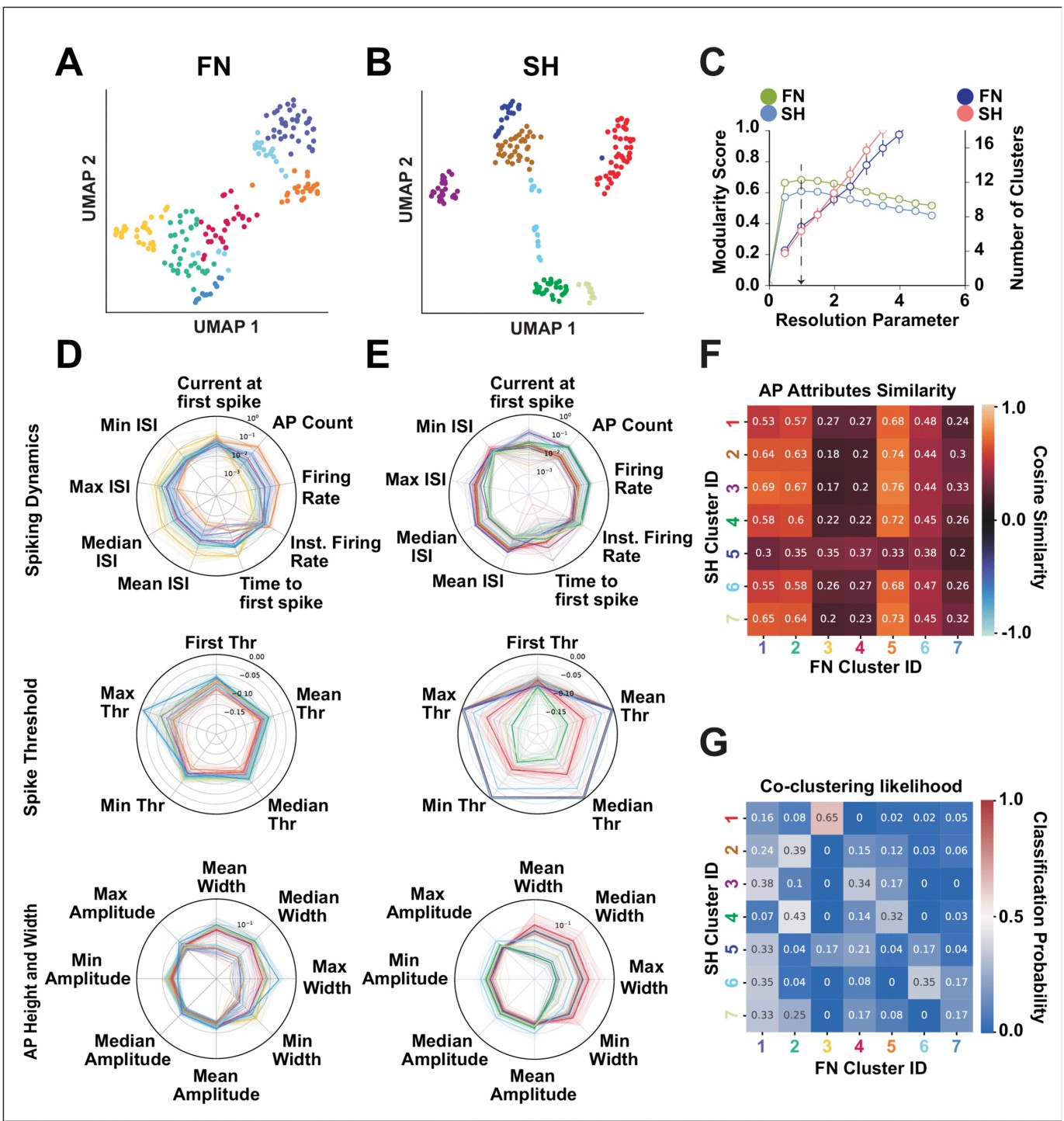

**Fig 2**. **Neural identity based on action potential (AP) attributes depends on stimulus type. (A-B)** UMAP projections of AP attributes for neurons recorded under FN and SH protocols, colored by cluster identity. **(C)** Cluster stability analysis showing mean ± SD modularity (left axis) and number of clusters (right axis) across resolution parameters (0-5, step 0.5), based on 25 iterations using 80% random subsampling. The black arrow marks the chosen resolution maximizing modularity. **(D-E)** Radar plots showing AP attributes grouped by spiking dynamics, spike threshold, and AP height/width features for SH and FN protocols. Thin lines represent individual trials; thick lines show cluster means. **(F)** Cosine-similarity heatmap comparing AP attributes between SH and FN clusters, indicating no high-similarity pairs (>0.8). **(G)** Likelihood heatmap showing SH-FN cluster correspondence. Low ARI (0.149) and AMI (0.206) scores indicate inconsistent class identity across input protocols.

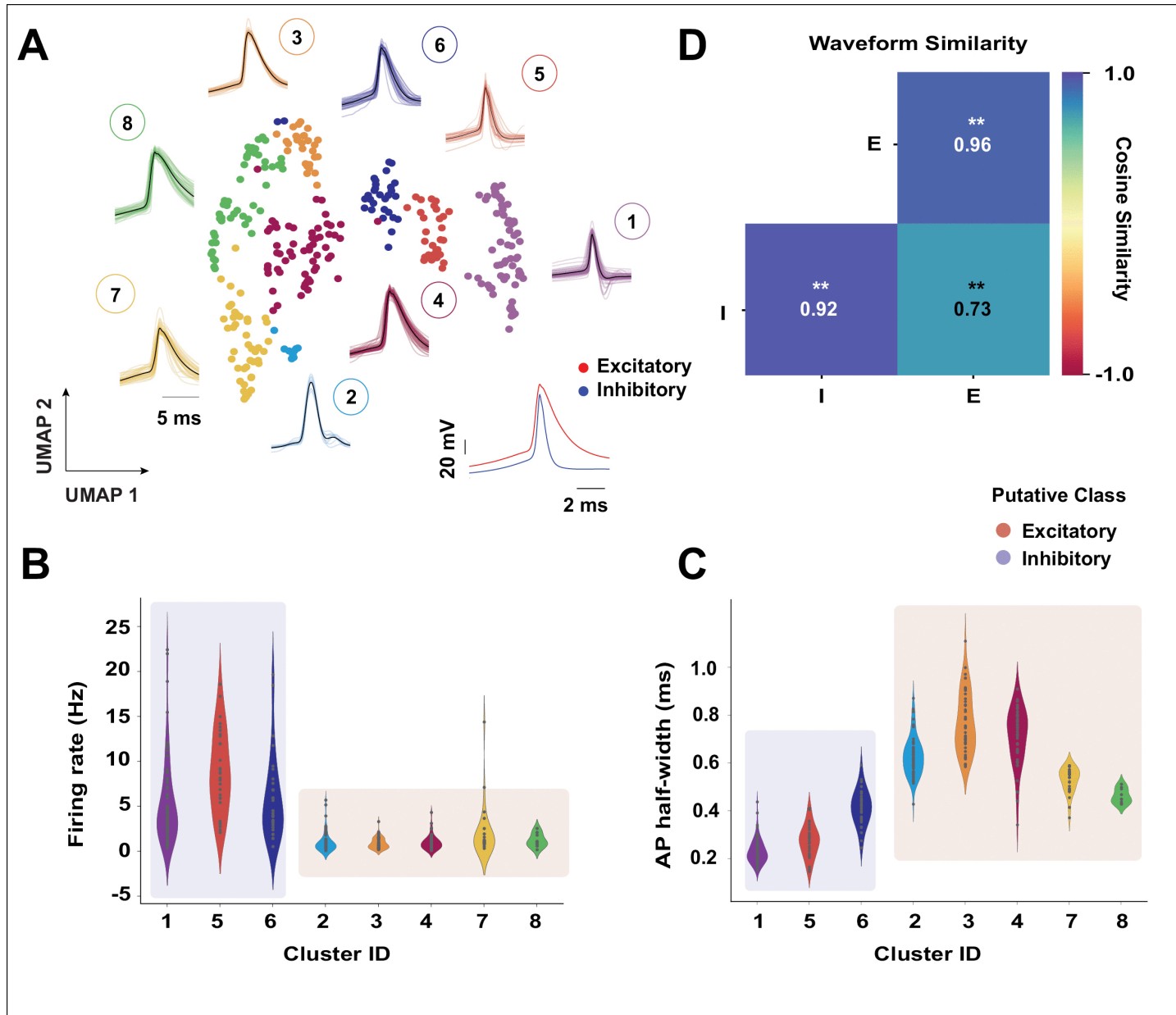

**Fig 3**. **Putative classification of excitatory and inhibitory neurons based on waveform and firing characteristics.** A total of 312 cortical neurons were recorded under the FN protocol. **(A)** UMAP projection of average waveforms from layer 2/3 neurons in the somatosensory cortex, colored by cluster identity; inset shows mean waveforms for putative excitatory and inhibitory groups. **(B)** Violin plots showing firing-rate distributions across clusters in (a); clusters 1, 5, and 6 ($n = 66, 30, 30$) exhibited higher firing rates and were labeled as putatively inhibitory ($n = 126$), whereas the remaining clusters ($n = 186$) were classified as putatively excitatory. **(C)** AP half-widths across clusters, with clusters 2, 3, 4, 7, and 8 ($n = 58, 48, 44, 27, 9$) showing lower values. **(D)** Average cosine similarity within and across excitatory (E) and inhibitory (I) waveform groups. Excitatory waveforms showed greater heterogeneity, and E/I populations differed significantly (Welch's ANOVA: $F(2, 49135) = 19148.00$, ***$p < 0.001$; Games-Howell post hoc: E vs I, I vs $E \times I$, and E vs $E \times I$, all **$p = 0.001$).

of the firing rates and half-widths for each cluster with matching colors using a violin plot (Fig 3B and 3C). We observe clusters 1,5 and 6 to have a narrow width and high firing rate relative to the rest of the clusters; therefore, we categorize these clusters as putatively inhibitory (I) and the rest as putatively excitatory (E). It is important to address that this is a putative classification and is prone to miss attributions without specific genetic markers. We compare the average cosine similarity between E/I populations. We find that the excitatory and inhibitory populations are significantly different (Welch's ANOVA, $F(2,49135) = 19148.00$, $p = 0.0$; Post-hoc Games-Howell test, E vs I ($p = 0.001$), I vs $E \times I$ ($p = 0.001$), and E vs $E \times I$ ($p = 0.001$), Fig 3D), but the excitatory population is more heterogeneous than the inhibitory one, based on the average similarity of the waveforms within the E/I population (Fig 3D).

**Action potential attributes based neuron profiles using the FN protocol.** As pointed out in the previous section, we aimed to understand and compare the usefulness of commonly used physiological attributes in uncovering functional classification in neuronal populations when neurons receive a physiologically realistic FN input. For that aim, we study commonly used action potential attributes for excitatory and inhibitory populations separately to discern if action potential attributes sufficiently capture within-population heterogeneity in excitatory and inhibitory groups. We also aim to unravel the differences between excitatory and inhibitory populations regarding their action potential attributes. For this, we extract 22 attributes (see Methods) subdivided into spiking dynamics, Spike threshold, and action potential height and width attributes with their descriptive statistics incorporating the mean, median, minimum, and maximum values.

We cluster the E/I populations separately based on the features and find 7 clusters for excitatory cells (shaded in red) and 6 clusters for inhibitory cells (shaded in blue). The UMAP representation of the spike-based properties for E/I populations with unique colors for each cluster for the chosen cluster parameter is shown in (Fig 4A and 4B). We show the cluster stability (see Methods) in (Fig 4C), the number of clusters is stable (low standard deviation) for the chosen resolution parameter (black arrow). We also test the stability of the clusters by excluding one attribute at a time and repeating the stability analysis (see Methods) and find the inhibitory clusters to be more stable to attribute exclusion (S3A and S3B Fig). We compare the means of action potential attributes simultaneously between excitatory and inhibitory populations and find them to be significantly different (one-sided MANOVA (see Methods), Wilks' lambda; $F(21, 290.0) = 35.1841$; $p = 0.000$, Pillai's trace; $F(21, 290.0) = 35.1841$; $p = 0.000$, Hotelling-Lawley trace; $F(21, 290.0) = 35.1841$; $p = 0.000$, Roy's greatest root; $F(21, 290.0) = 35.1841$; $p = 0.000$). To identify the relative importance of each attribute in the separation of excitatory and inhibitory populations, we perform a canonical correlation analysis (CCA) (see Methods) between the excitatory and inhibitory action potential attributes and calculate the loading (structure correlation) for each attribute (Fig 4D). We find that spiking dynamics attributes have the most influence on the inhibitory canonical variates suggesting that these variables have the most influence on separating the inhibitory population apart from the excitatory population. Alternatively, AP height and width attributes strongly influence the excitatory population canonical variate, suggesting that AP height and width are the most discriminatory. These results also show that individual action potential attributes contribute differently to the canonical variate (i.e., the latent structure) for excitatory and inhibitory populations respectively.

Next, we want to compare the level of heterogeneity within excitatory (E-E) and inhibitory (I-I) populations as well as across (I-E) populations using the action potential attributes. We visualize the action potential attributes using a radar plot shown in (Fig 4E and 4F). It shows the diversity of spiking dynamics, spike threshold, and AP height and width for each neuron in each cluster (with the same color as in the UMAP projection (Fig 4A and 4B)) along with the mean for the cluster (thick line). We observe that spiking dynamics attributes differ between excitatory and inhibitory clusters. The spiking threshold profiles were also found to be different across E/I clusters. Comparing the AP height and width attributes, however, show a similar profile across E/I clusters. We quantify the heterogeneity in the E/I clusters using a cosine similarity measure. The excitatory population had a significantly higher cosine similarity measure within the population than the inhibitory population (Welch's ANOVA, F(2,49135)=3626998.65, p= 0.0; Post-hoc Games-Howell test, E-E vs I-I (p=0.001), I-I vs I-E (p=0.001), and E-E vs I-E (p=0.001), (Fig 4G)), suggesting that inhibitory action potential attributes are more heterogeneous than their excitatory counterparts. The mean cosine similarity score between excitatory

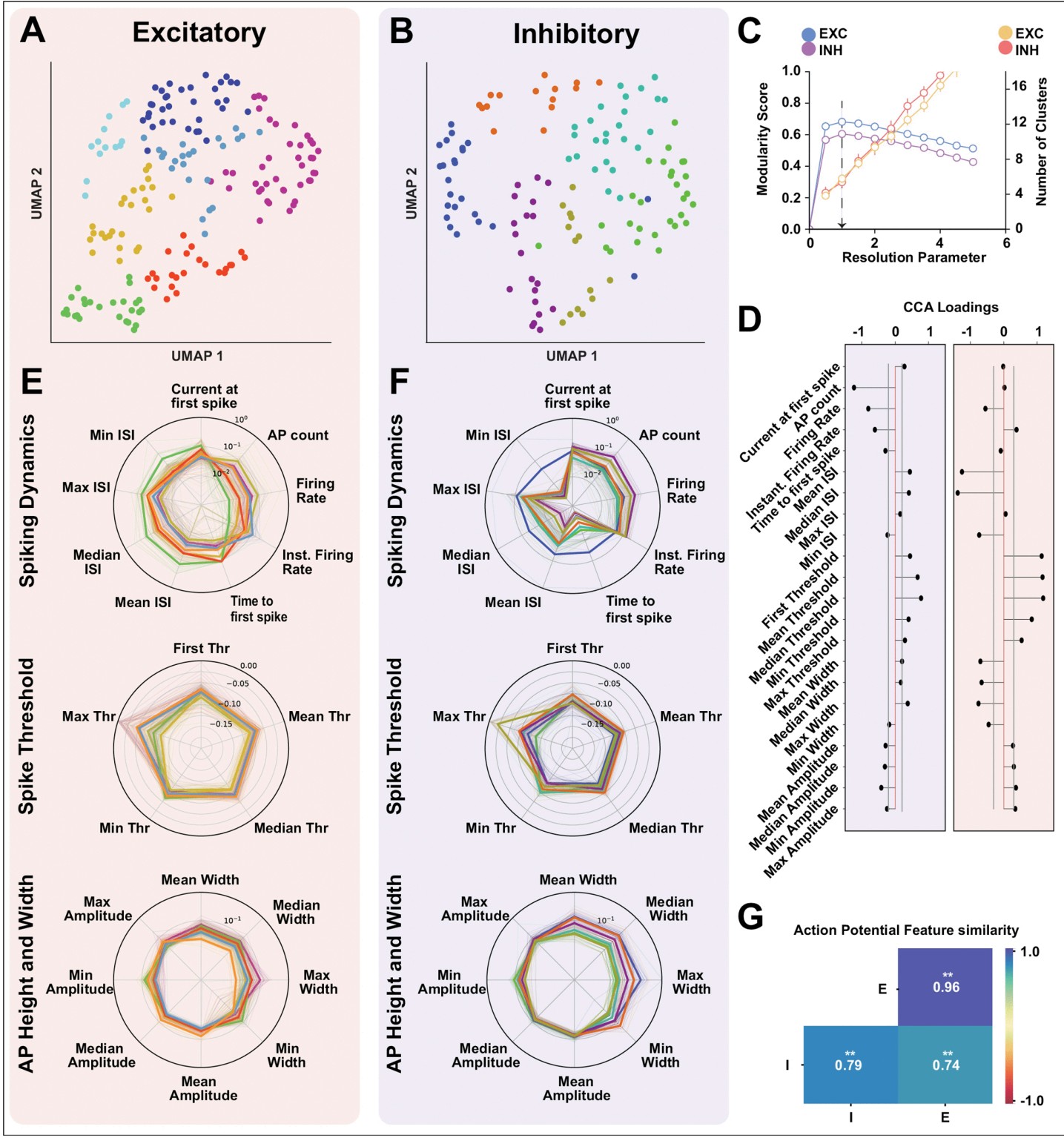

**Fig 4**. **Cell type-specific heterogeneity and the principal discriminatory attributes of the action potential.** Putative excitatory (red background) and inhibitory (blue background) neurons were analyzed separately. **(A-B)** UMAP-based clustering grouped excitatory and inhibitory populations into 7 and 6 clusters, respectively. **(C)** Chosen resolution parameters corresponding to maximal modularity and resulting cluster counts. **(D)** Multivariate analysis confirmed significant differences in AP attributes between excitatory and inhibitory populations. (one-sided MANOVA; Wilks' lambda, Pillai's trace,

Hotelling-Lawley trace, and Roy's greatest root: $F_{(21, 290)} = 35.18$, ***$p < 0.001$). Post hoc canonical correlation analysis (CCA) shows attribute loadings with 95% confidence intervals (grey lines). **(E-F)** Radar plots of AP attributes grouped by spiking dynamics, spike threshold, and AP height/width. Thick lines show cluster means; thin lines represent individual neurons. Spiking-dynamics parameters exhibit the largest variability across clusters. **(G)** Cosine similarity between excitatory-excitatory (E-E), inhibitory-inhibitory (I-I), and cross-type (I-E) feature vectors. Inhibitory neurons display greater heterogeneity in AP attributes (Welch's ANOVA: $F_{(2, 49135)} = 3,626,998.65$, $p < 0.001$; Games-Howell post hoc: all pairwise comparisons **$p = 0.001$).

and inhibitory populations are significantly lower than the within-population cosine similarity score for both excitatory and inhibitory populations respectively, suggesting that excitatory and inhibitory feature action potential attribute vectors are different from each other, reiterating the MANOVA results. The results above demonstrate that action potential attributes are different between excitatory and inhibitory populations. The inhibitory population is more heterogeneous in its action potential attributes than the excitatory population.

**Passive biophysical feature and adaptation-based profile using the FN protocol.** As we describe in the last two sections, we aim to compare neuronal attribute sets to find the attributes(s) most informative about heterogeneity in excitatory and inhibitory populations and try to understand how neurons cluster based on these properties. The passive biophysical attributes (i.e., membrane resistance, capacitance, etc.) shape the response properties of a neuron. Moreover, it has been reported that adaptation current which captures the adaptation properties of the membrane is another important neuron property that can determine cell classes with noisy input [29]. We, therefore, consider the passive biophysical attributes and the adaptation current as potential candidates that capture heterogeneity within the neuronal population. To understand the heterogeneity of passive biophysical properties and adaptation parameters across the excitatory and inhibitory populations, we extract both a set of passive parameters (see methods) and an adaptation current by fitting a Generalized Leaky Integrate and Fire (GLIF) model on the first 100 seconds of each FN trial, using the automated method described by [27] (see methods). (Fig 5A) shows an example of a 10s instance of a GLIF-fitted model and one of the original recordings. We characterized the goodness of fit by measuring the explained variance (EV) between the subthreshold membrane potential of the original data and the model. (Fig 5B) shows the distributions of the EV and $\Gamma$ for the entire dataset. We eliminate the models with an EV value below 0.7 and obtain a set of 307 samples (out of 312 samples).

We cluster the recordings into cell classes based on a set of passive biophysical parameters along with threshold adaptation constants (in total 6 features, see Fig 5A and 5B) using the UMAP+Louvain clustering method explained in the previous sections, we find 7 clusters for the excitatory population and 6 clusters for the inhibitory population. We examine the stability of the clusters (Fig 5C) with different hyper-parameters (resolution parameters, see Methods) and choose (black arrow) the resolution parameter with the maximum modularity score. The clusters for the chosen hyperparameter for both excitatory and inhibitory populations were stable (low standard deviation). We also test the stability of the clusters by excluding one attribute at a time and repeating the stability analysis (see Methods) and found the inhibitory clusters to be more stable to attribute exclusion (S3C and S3D Fig). (Fig 5D and 5E) shows the UMAP representation of the feature set (inset) for excitatory (red background) and inhibitory (blue background) populations, the radar plot shows the individual passive biophysical parameter values for each neuron and each thin line in the radar plot represents a single neuron and are color matched to respective clusters, the mean for each cluster is plotted with a thick line. We compare the means of all the passive biophysical attributes simultaneously between excitatory and inhibitory populations and find them to be significantly different (one-sided MANOVA, Wilks' lambda; $F_{(6, 301)} = 23.2629$; $p = 0.000$, Pillai's trace; $F_{(6, 301)} = 23.2629$; $p = 0.000$, Hotelling-Lawley trace; $F_{(6, 301)} = 23.2629$; $p = 0.000$, Roy's greatest root; $F_{(6, 301)} = 23.2629$; $p = 0.000$). To identify the relative importance of each passive biophysical attribute in the separation between excitatory and inhibitory populations, we perform a CCA analysis (see Methods) between the excitatory and inhibitory passive biophysical attributes and calculate the loading (structure correlation) for each attribute (Fig 5H). We find dissimilar contributions from excitatory and inhibitory passive parameters towards their respective canonical variate,

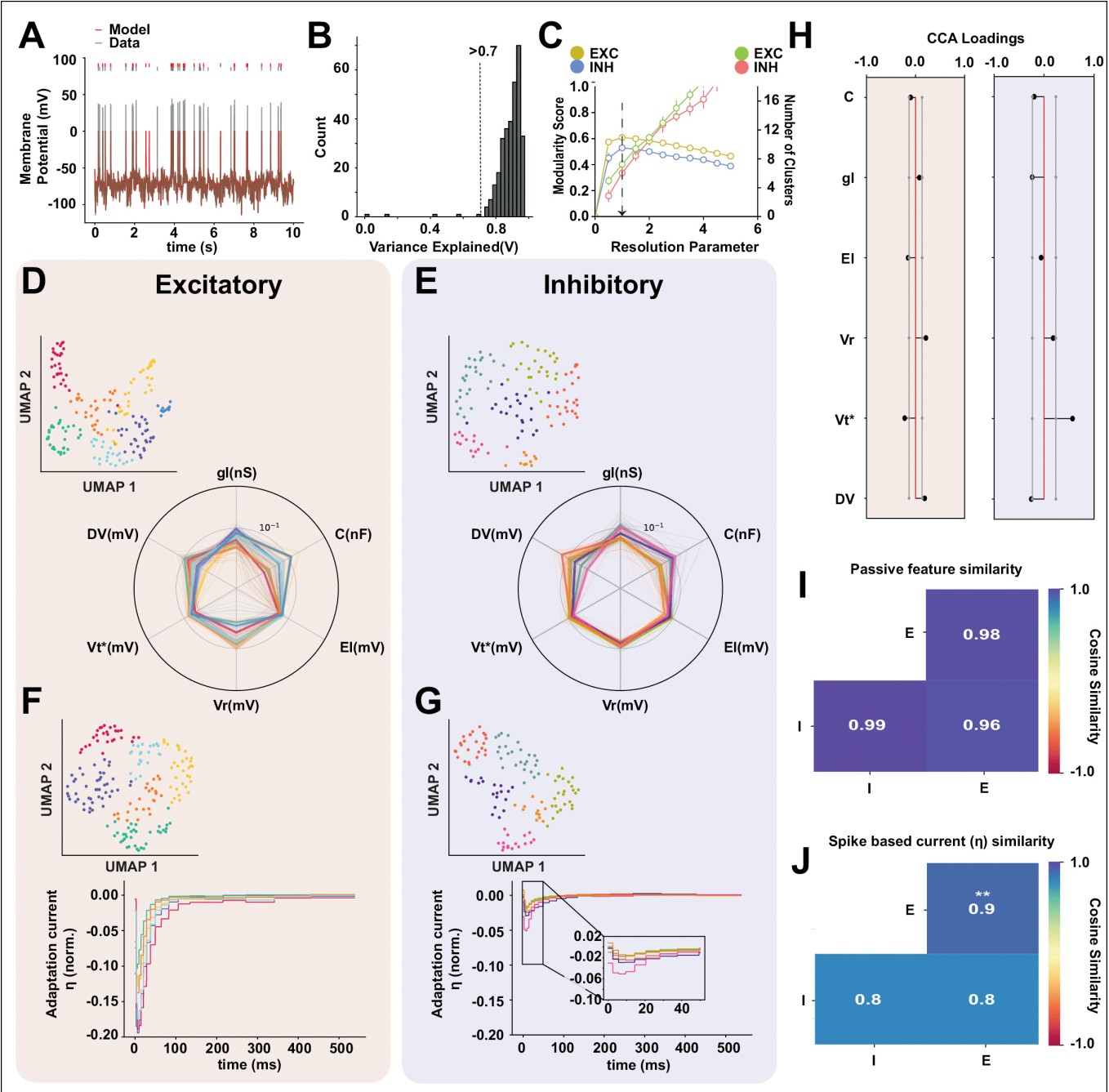

**Fig 5**. **Discriminatory power of passive biophysical and adaptive membrane properties for neuronal classification.** A GLIF model was fitted to FN-stimulated membrane potential recordings. **(A)** Example 10-s snippet showing model output (red) and experimental trace (gray). **(B)** Histogram of subthreshold variance explained; only models with $R^2 > 0.7$ (dashed line) were used for clustering. **(C)** Chosen resolution parameters yielding maximal modularity and corresponding cluster counts. **(D-E)** UMAP representation and radar plots of passive biophysical parameters for excitatory (7 clusters, left) and inhibitory (6 clusters, right) populations. **(f-g)** Same for adaptation current parameters. **(H)** Post hoc CCA shows attribute loadings with 95% confidence intervals. Excitatory and inhibitory passive parameters differed significantly (MANOVA; Wilks' lambda: $F_{(6, 301)} = 23.26$, ***$p < 0.001$; Pillai's trace, Hotelling-Lawley trace, and Roy's root identical). **(I)** Cosine similarity among E-E, I-I, and I-E passive attribute vectors showed no significant difference (Welch's ANOVA: $F_{(2, 66792)} = 0.17$, $p = 0.84$). **(J)** Adaptation current similarity differed significantly across groups (Welch's ANOVA: $F_{(2, 46968)} = 2796.77$, **$p < 0.001$; Games-Howell: I-I vs E-E **$p = 0.001$, I-I vs I-E $p = 0.75$, E-E vs I-E **$p = 0.001$), indicating greater inhibitory heterogeneity.

suggesting that each passive biophysical attribute contributes differently to the excitatory-inhibitory latent structure and therefore sets the excitatory-inhibitory populations apart. Next, we compare the level of heterogeneity between the passive feature vectors across the E/I populations (Fig 5I) using the cosine similarity measure within and across excitatory populations and find no significant difference between the mean cosine similarity measure within excitatory and inhibitory populations as well as across excitatory-inhibitory population (due to low effect size) (Welch's ANOVA, F(2,66792) = 0.16777; p= 0.84, Fig 5I). These results suggest that even though the mean of the excitatory and inhibitory passive biophysical attributes are significantly different from each other (measured using MANOVA), the level of heterogeneity within excitatory-inhibitory populations respectively based on passive biophysical parameters is quite low. Therefore the passive biophysical attributes are not so informative about the neuronal heterogeneity.

As we saw above, passive biophysical attributes are not informative about population heterogeneity, this result is consistent with [29], which finds that passive biophysical attributes do not distinguish between neuron types. The adaptation current is more useful in putting neuron types apart. Therefore we classify the neurons using the adaptation current ($\eta$) extracted from the fitted GLIF model for the E/I population separately. We find 6 classes for the excitatory population (red) and 5 classes for the inhibitory population (blue). In (Fig 5F and 5G) we show the UMAP representation of the adaptation current and the corresponding average normalized shapes for each cluster (inset) each cluster has its respective color in the UMAP as well as shape plots. We observe that the excitatory adaptation currents have a stronger negative amplitude than their inhibitory counterparts. Moreover, the adaptation currents of the inhibitory population relax back to their resting values at earlier times than for the excitatory population.

We quantify the heterogeneity of adaptation currents within excitatory and inhibitory populations using the cosine similarity measure. The heatmap in (Fig 5J) shows that the excitatory adaptation currents have a significantly higher average cosine similarity compared to the inhibitory adaptation currents, suggesting that the inhibitory adaptation current is more heterogeneous than the excitatory population (Welch's ANOVA; F(2,46968) = 2796.77, p = 0.0; Post-hoc Games-Howell test E vs I (p = 0.001), I vs $E \times I$ (p = 0.75), and E vs $E \times I$ (p = 0.001), Fig 5J). The similarity measure is not significantly different between the inhibitory (I) population and across the inhibitory and excitatory (I vs E) population. This suggests that inhibitory adaptation currents are as different from each other as they are different from the excitatory adaptation currents. These results suggest that adaptation current profiles are different between excitatory and inhibitory populations. These results also show that adaptation currents are useful for understanding the heterogeneity within the inhibitory population but not so much for the excitatory populations.

**Neuronal classification based on linear input filter approximated using a Spike Triggered Average (STA) in the FN protocol.** In last two sections, we explored neuronal heterogeneity based on action potential, passive biophysical, and adaptation attributes in neurons while responding to an FN input. We found that except for passive biophysical attributes, other properties show both within population across population heterogeneity for both excitatory and inhibitory populations. We observe a higher level of heterogeneity for inhibitory populations. Since we aim at understanding the extent to which various neuronal properties help uncover neuronal diversity, in this section we focus on the linear input filter of neurons and aim to understand the neuronal diversity based on this attribute. We consider this attribute important as neurons respond strongly to specific features in the input [49], making linear input filter an important property to study the functional diversity of the neuronal population.

The STA, as described in [30], estimates a neuron's linear input filter by identifying the features in the input that trigger the neuron to spike. It does this by averaging the input signal over a specific time window preceding each spike, using data from all observed spikes. This makes the STA a useful method to approximate the linear input filter of a neuron. We want to investigate the STA diversity across the E/I population, for this, we extract the STA from all the neurons (see Methods). For calculating the STA, we use the injected current which is the result of shifting a theoretical dimensionless input with a constant baseline and scaling it by a factor as explained by [17,18].

We cluster the excitatory and inhibitory population STAs separately using an unsupervised UMAP+Louvain clustering method (see Methods). We find 7 clusters for the excitatory population STAs (red) and 8 for the inhibitory population STAs. The STAs were normalized and standardized before clustering (see Methods). The UMAP representation and the corresponding averaged and normalized STAs for each cluster are shown in (Fig 6A and 6B). A visual inspection shows that inhibitory and excitatory STA shapes differ in their peak amplitudes and in the maximum slope of their initial rise (Fig 6A and 6B inset), this difference is shown more clearly in (Fig 6D). The stability of the clustering algorithm for the chosen parameters is shown in (Fig 6C). To quantify the heterogeneity within the excitatory and inhibitory population as a result of that linear input filters, we first calculate the STA shape similarity between clusters, using a cosine similarity

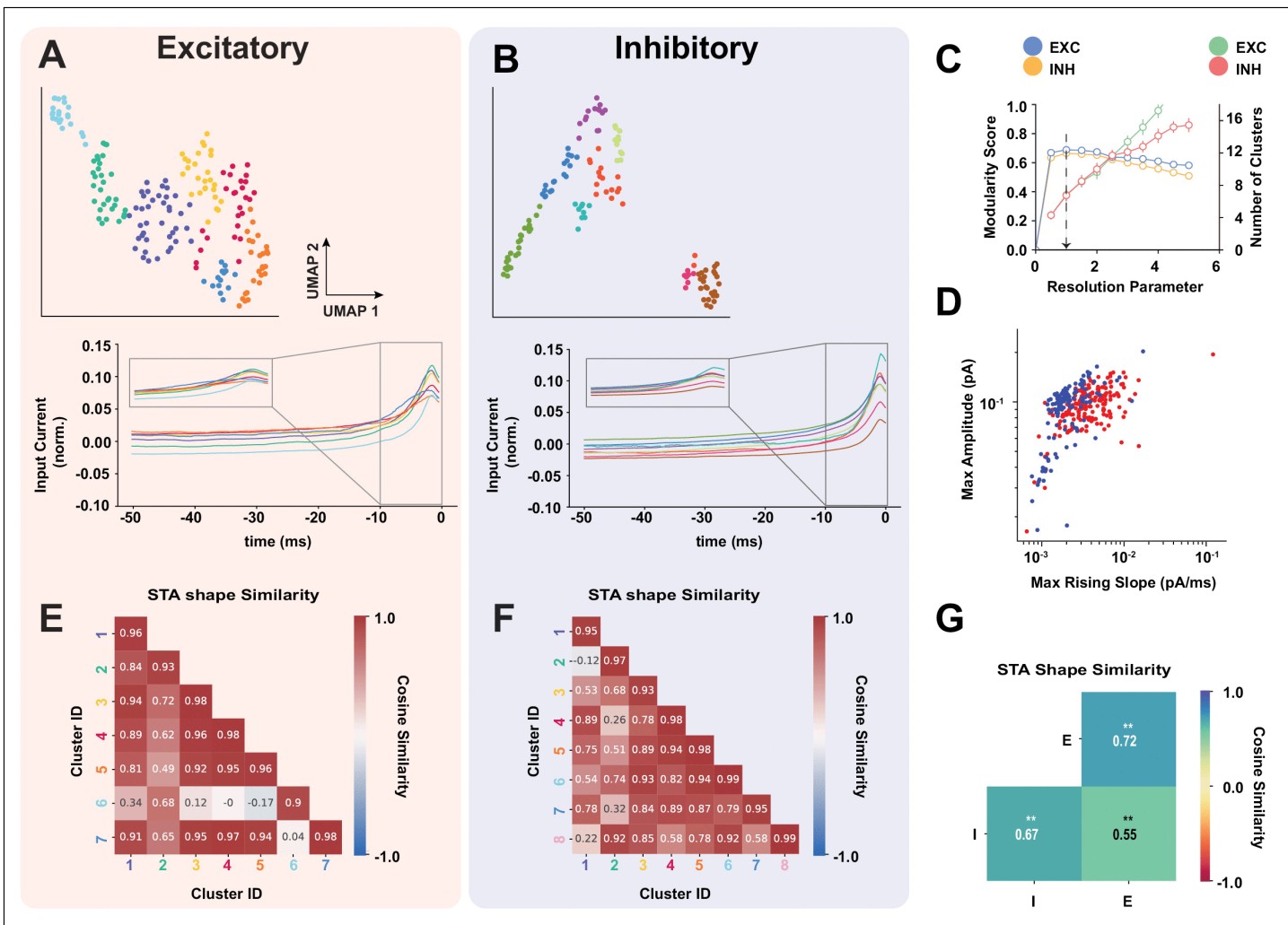

**Fig 6. Linear Input Filter measured using a Spike Triggered Average (STA) (A-B)** UMAP representations of excitatory (left, 7 clusters) and inhibitory (right, 8 clusters) populations with corresponding normalized average STA shapes in the 50 ms preceding a spike. **(C)** Resolution parameters chosen for maximal modularity and resulting cluster counts. **(D)** Scatter plot of maximum slope versus maximum amplitude for each STA (excitatory in red, inhibitory in blue). **(E-F)** Average cosine similarity within excitatory (e) and inhibitory (f) STA clusters. Excitatory clusters exhibit higher intra-group similarity (>0.9 with ~2 other clusters on average), whereas inhibitory clusters show similarity with only ~1 other cluster. **(G)** Heatmap of average cosine similarity within (E-E, I-I) and across (E-I) populations. Inhibitory neurons display greater heterogeneity (lower similarity). Welch's ANOVA: $F(2, 48825) = 1005.90$, ***$p < 0.001$; Games-Howell post hoc: E vs I **$p = 0.001$, I vs E-I **$p = 0.001$, E vs E-I **$p = 0.001$.

between excitatory and inhibitory STA clusters (Fig 6E and 6F). The values shown in the heatmap in Fig 6E and 6D were calculated by averaging the cosine similarity matrices between STAs of each cluster pair and within each cluster (for the within-cluster comparison, only the upper triangular values were included in the average). The average cosine similarity value between excitatory STA clusters shows higher values (>0.9) with each other except for clusters IDs 2 and 6, which show a low similarity with every other cluster except for itself. On average, each excitatory cluster shows a high similarity (>0.9) with more than 2 other clusters. On the other hand, the average cosine similarity values between inhibitory clusters is low (<0.9), with some exceptions such as cluster IDs (2, 8), (3, 6), (5, 4), (6, 5) and (8, 6). On average, the STA of each inhibitory cluster is highly similar to approximately 1 other cluster. The average cosine similarity value is high between STAs within each inhibitory cluster. We summarize the STA similarities within the excitatory (E-E) and inhibitory populations (I-I) as well as across the two populations (I-E) in the heatmap in (Fig 6G). This is done by averaging the cosine similarity matrices between STAs within excitatory (E-E) and inhibitory (I-I) populations as well as across excitatory and inhibitory (I-E) populations, these matrices are shown in (S5B Fig). For within-population averages (E-E and I-I), we only take the upper-triangular portion of the matrix. The average cosine similarity value within the inhibitory population is significantly lower than that of within the excitatory population (Fig 6G Welch's ANOVA, $F(2,48825) = 1005.90$, $p = 0.0$; Post-hoc Games-Howell test, E vs I ($p = 0.001$), I vs $E \times I$ ($p = 0.001$), and E vs $E \times I$ ($p = 0.001$)), suggesting that inhibitory STAs are more heterogeneous than excitatory ones. Also, the STA shape similarity between excitatory and inhibitory populations is significantly lower than the within excitatory and inhibitory similarity, suggesting that excitatory and inhibitory STAs are different. We can see that on average inhibitory neurons respond to more diverse features in the input than excitatory neurons.

**Comparing physiological heterogeneity across attribute sets using multi-set correlation and factor analysis.** We aim to explore what a physiologically realistic stimulus reveals about the functional heterogeneity of a neural population. For this aim, we need to compare FN-based neuronal attribute sets and find out which attribute set is the most informative about neuronal heterogeneity under the dynamic FN stimulation protocol. This is a complicated problem, as the four attribute sets are of varying dimensions. To perform this comparison, we use a method known as Multi-set Correlation and Factor Analysis [40], which is an unsupervised multi-set integration method based on probabilistic principal Component Analysis (pPCA) and Factor Analysis (FA), that can help to understand the common and shared factors across multi-modal data (see methods & [40]). We use this method to compare the private and shared variance explained by the 4 attribute sets we examined in the previous sections (i.e., action potential attributes, passive biophysical attributes, adaptation currents, and linear input filters (STA)). We use the pPCA space to model the shared structure across attributes (see Methods), and the residuals are modeled as a private structure for each attribute set using factor analysis (see methods).

Comparing the variance explained by the shared structure across attributes, we found that the excitatory population which is inferred by a 4-dimensional shared structure, explains almost 50% of the variance in action potential and passive biophysical attributes (Fig 7A top), this is higher than linear input filter (STA) (32.9%) and adaptation current attribute (19.7%). Most importantly, we see that the linear input filters (STAs) explain the highest private variance (57.2%), followed by adaptation current (41.2%). The action potential and passive biophysical attributes have a relatively lower private variance (35.7%). We can further investigate the contribution to the most important shared factor by each attribute, through which we find that action potential and passive biophysical attributes explain the most variance for the most important shared dimension (Fig 7A (middle, bottom)). Similarly, for the inhibitory population, explained by a 3 dimensional shared structure, we find that the linear input filter (STA) explains the most amount of private variance (80.6%), much higher than the excitatory population (Fig 7B (top)), followed by adaptation current (29.4%), action potential attributes (35.7%), and passive biophysical attributes (17.1%) respectively. The action potential attributes and passive biophysical attributes explain the most and almost equal amount of shared variance (33.8% and 33. 85% respectively) (Fig 7B (middle)), followed by adaptation current (29.4%) and linear input filter (9.9%) (Fig 7B (top)). We find that the

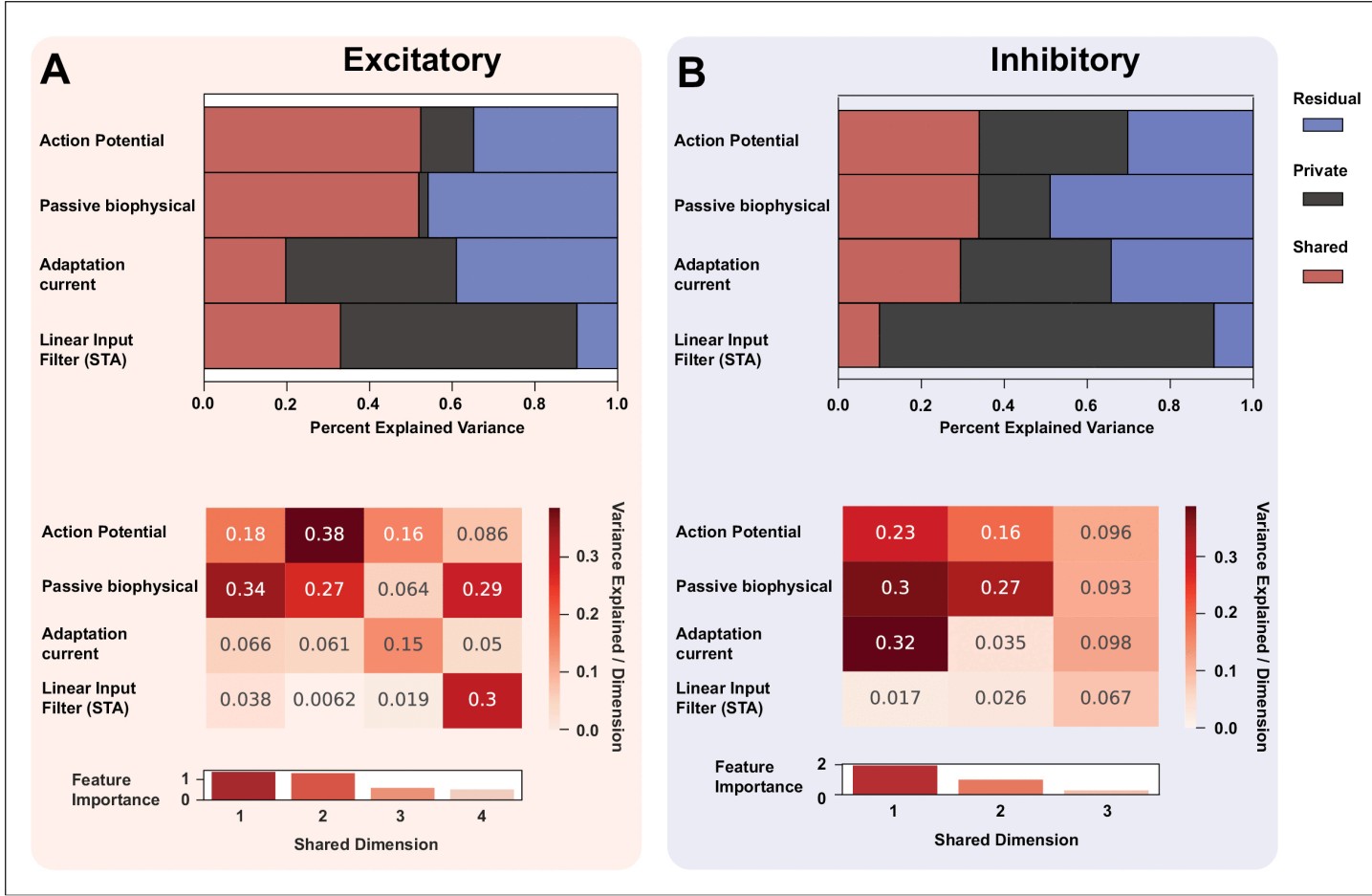

**Fig 7**. **MCFA analysis comparing shared and private variance explained by Action potential, Passive biophysical, Adaptation current, and Linear input filter (STA)**: **(A) Excitatory population:** Histogram shows variance explained by each attribute. Action potential (AP) and passive bio-physical parameters account for most shared variance (AP: shared 52.3%, private 12.7%; passive: shared 51.9%, private 2.2%), indicating low unique informativeness. STA captures most unique variance (shared 32.9%, private 57.2%) with minimal residual, while adaptation current has intermediate private variance (private 41.2%). Heatmap shows contributions of four shared dimensions, with AP and passive parameters dominating. **(B) Inhibitory population:** AP and passive parameters explain the most shared variance (AP: shared 33.9%, private 35.7%; passive: shared 33.85%, private 17.1%), though passive parameters also have high residual. STA explains most private variance (private 80.6%) with low residual, followed by adaptation current (private 29.4%). Heatmap of three shared dimensions shows contributions from AP, passive, and adaptation currents. Overall, STA is the strongest marker of population heterogeneity, highlighting the role of input in distinguishing excitatory and inhibitory neurons.

adaptation current and passive features explain the largest amount of variance for the most important shared dimension (Fig 7B (bottom)) for the inhibitory population. It is important to observe that the linear input filter explains more than 85% of the total variances (shared+private) for both excitatory and inhibitory populations, higher than the other 3 attribute sets.

The high values of private variance explained by the STAs in both excitatory and inhibitory cases show that passive biophysical, action potential, and adaptation currents are not highly correlated with the linear input filter of a neuron. Therefore, the linear input filter contains unique information about the excitatory and inhibitory population that is not shared with the other 3 attribute sets. Most importantly, the high value of private variance for the linear input filter, along

with low residual values compared to other attribute sets indicates that linear input filters are likely to be the most informative attributes for explaining neuronal heterogeneity compared to passive biophysical, action potential, and adaptation current attributes. On the contrary, high values of shared variance for passive biophysical attributes and action potential attributes show that these attributes share a common structure, and can be a good predictor of one another. It is also important to observe that the total variance explained (shared + private) by adaptation currents is lower than action potential attributes for both excitatory and inhibitory populations, suggesting that action potential attributes are more informative about neuronal heterogeneity than adaptation currents.

## Discussion

In this work, we aimed to study the effect of input protocol (SH vs FN) on neuronal functional classification as well as to draw consensus on the attribute that are most informative about functional heterogeneity. For this, we first looked at the effect of the input protocol on neuronal classification based on two feature sets: waveforms and action potential attributes, while neurons were stimulated using two stimulus protocols: a static SH and a physiologically realistic FN input. We found neuronal classification based on FN and SH protocols to be inconsistent when comparing the similarity in the cluster assignment in both input conditions using the co-cluster likelihood and cluster similarity indices (ARI and AMI). This highlights the importance of using a physiologically realistic input for studying functional heterogeneity of a neuronal population. We then aimed to determine which attribute(s) is/are the most informative about the neuronal functional heterogeneity when neurons are stimulated with a physiologically realistic FN input. We first explored the functional diversity of cells using 4 attribute sets (action potential, passive biophysical, adaptation current, and linear input filter (STA) attributes) for putative excitatory and inhibitory populations. To infer which attribute(set) is most informative about neuronal heterogeneity, we compared the private and shared variance across the four attribute sets. We found that the linear input filters (STAs) explain the most variance, especially private variance across the excitatory and inhibitory neurons, and thus, contains unique information about neuronal functional heterogeneity compared to other attributes and is the most informative about neuronal functional heterogeneity.

### The role of input in neuronal functional diversity

In the present work, we found that neuronal functional classification changes as a function of the input the neurons receive, this point is still not discussed in neuronal clustering literature. Our results fill an important gap in the literature, the effect of input on physiological classification, by showing that waveforms and electrophysiological attributes-based classifications are different when neurons are stimulated with static versus dynamic inputs. Previous studies that combined electrophysiological, morphological, and molecular attributes for neuronal classification [9,11,14], hypothesized that electrophysiological classification of a neuron can potentially change when using a different stimulus protocol. Our classification results confirm this hypothesis, by showing that clustering based on intracellular waveforms and action potential attributes are inconsistent across SH and FN-based inputs.

Our results showcase the importance of considering the functional dimension of neuronal identity that emerges from the interaction between input characteristics and intrinsic neuronal properties. These results do not only showcase how neurons cluster differently as a result of changing input protocol, which is an important and novel finding in the context of neuronal classification, but they also show that spiking dynamics, spike threshold, and AP height and width attributes were distinct between FN and SH protocols, as shown by the consistently low cosine similarities between FN and SH clusters and the radar plots comparing the spiking dynamics, spike threshold, and AP height and width attributes for each neuron. This is in agreement with earlier results of [42,43,50], which have already shown that firing intensity measures (AP count and AMPA conductance) are weakly correlated between static and dynamic stimuli (dynamic clamp) conditions. We extended these results by showing that the commonly studied action potential attributes span different latent manifolds as a result of input protocol (S1 Fig). Therefore, these classical action potential attributes do not encode a signature

of the neurons that is invariant to the input, but are susceptible to the input the neurons receive. Since the conventionally used step and hold protocol does not represent an input a neuron receives in vivo, our findings advocate for using a physiologically realistic dynamic inputs such as the FN input for studying functional diversity. Our results therefore establish that input dynamics and its effects on functional classification and need to be considered even before molecular and morphological markers.

We used an unsupervised UMAP+Louvain clustering method in this study which has already been shown to distinguish neuronal classes based on extracellular waveforms [35] and also has been shown to improve upon classifications based on a low number of features extracted from waveforms. We capitalized on this idea by using UMAP+Louvain clustering to cluster SH and FN intracellular waveforms and action potential attributes and then compared the cluster assignments across the two protocols. Since the Louvain community method clusters on the high-dimensional graph structure provided by the UMAP algorithm, rather than projecting the data on a low dimensional space [9,51], which often leads to a loss of information about the latent structure [51], the clusters using the UMAP graphs were more robust than found either using a dimensional feature set or a low dimensional projection of the original high dimensional feature set. A second important advantage of using a non-linear dimensionality reduction technique like UMAP is that it allows for a manifold comparison between the two input conditions (see S1A and S1B Fig).

### Neuronal classification using frozen noise input-based attributes

We have established that neuronal functional classification is a result of the type of input that neurons receive, therefore a physiologically realistic input is important to understand neuronal functional diversity. A new question emerges: which attribute sets are the most informative about neuronal functional heterogeneity when neurons are presented with a dynamic stimuli? A similar question has been raised by [8]. We attempted to provide a schema for answering this question. We did not intend to suggest a definitive number of classes of neurons, nor did we want to match our findings with previously established MET type classification, but we rather aimed to delineate a framework for drawing a consensus about which attribute(s) are the most informative about neuronal functional heterogeneity. An array of previous classification studies have relied upon variants of action potential and passive biophysical attributes [52–55], without any consensus about the informativeness of the attributes selected. Also, these studies have conventionally been reliant on feature extraction based on step stimulus protocols, which we demonstrate to produce cell classes that are different from when the neurons are stimulated with a dynamic input.

Although [42,43] provide a comparison between neurons responding to static and dynamic inputs, an active recommendation for which parameters are the most discriminatory was still missing. We supersede this by offering an alternative classification paradigm, based on the physiologically realistic frozen noise input. We tried to fill this gap by dividing the neuronal population into putative excitatory and inhibitory groups, then clustering neurons using the unsupervised UMAP+Louvain method, using four different sets of attributes separately. We found that neuronal clusters are subjective to feature selection. There was little consistency in comparing cluster assignments across the four attribute sets (see S6A and S6B Fig): the number of classes as well as the cluster assignments were found to be inconsistent across the four attribute sets.

We divided the neuronal population into putative excitatory and inhibitory classes based on the waveform shapes and firing rates, which is rather different from using molecular and morphological labels for E/I classification. Still, the goal of this study is not to align the neuronal identities to their ME-type markers, but rather to highlight the importance of input stimuli in functional classification. We show this by demonstrating that the firing properties of the narrow-width and broad-width neurons, which are conventionally categorized into inhibitory and excitatory classes respectively, change as a result of changing input type.

Parallel to investigating how putative E/I populations cluster based on different features, we also estimated the within-population variance for each feature set within E/I populations, which is representative of population heterogeneity based on a said feature. For action potential and passive biophysical attribute sets, we calculated the differences in the means of cosine similarity between E/I populations. We found that action potential attributes significantly differed between excitatory and inhibitory populations, consistent with previous findings [15]. We also found that the action potential attributes for the inhibitory population was more heterogeneous than the excitatory population, which is also consistent with previous findings [9,11]. It is important to highlight that the number of clusters found in our analysis is similar across excitatory and inhibitory populations, which might result from our dataset containing more excitatory than inhibitory neurons. The clustering based on the second feature set, that of passive biophysical properties extracted by fitting a GLIF model to the neural recordings, showed that passive parameters were also significantly different between excitatory and inhibitory neurons, but within-population heterogeneity was not significantly different between excitatory and inhibitory neurons. This suggests that passive parameters do not drive functional heterogeneity. This result is consistent with [29], which suggests that passive properties are not sufficiently discriminatory within E and I populations. Moreover, the authors claim that adaptive properties extracted using the GLIF model, such as the adaptation current, are more discriminatory than passive parameters. Our findings confirm this: adaptation currents show low similarity between excitatory and inhibitory populations, we found that the absolute maximum amplitude of all the adaptation current of the inhibitory neurons is smaller than that of the excitatory ones. We also found that inhibitory neurons have a significantly more heterogeneous adaptation profile than the excitatory population. This result provides data for designing and studying heterogeneous adaptive network models to further enhance our understanding of neural circuits' functional underpinnings.

Neurons can be functionally classified based on their input response features [49,56,57]. We analyzed the linear input filters of neurons using the Spike Triggered Averages (STAs). This technique has been extensively used in studying the stimulus preference of neurons in the visual cortex [58,59]. Our results show that the STA features are effective for estimating neuronal functional heterogeneity. We found that on average the STA shapes between the excitatory and inhibitory populations were different, based on comparing the average cosine similarity values between E and I populations, showing the difference in linear input filter between the two populations. We also found that the STA heterogeneity within the inhibitory population is significantly higher than the excitatory population. It has been shown [60] that visual cortical fast-spiking (putatively inhibitory) and regular-spiking (putatively excitatory) neurons have distinct levels of feature selectivity due to differences in passive biophysical attributes, such as the membrane time constant and input resistance. Since fast-spiking neurons have higher membrane leak conductance, therefore lower resistance it results in sharpening of neuronal selectivity to its preferred input as shown in previous studies such as [61]. Conversely, regular spiking neurons were found to have lower conductance and thus, lower sensitivity to preferred stimuli. Previous computational studies have found that physiological and passive biophysical have degenerative relationships with STA kernels [62], this study has shown that the same STA shape can be achieved by multiple difference values of the passive biophysical properties. This kind of degeneracy has been observed at the ion channel level as well [63,64]. On the contrary, we observed a limited variability in passive biophysical attributes giving rise to a higher heterogeneity (comparing the within-population similarity matrices) in the STAs, i.e. in the functional linear input filter. We expect that the measured passive biophysical parameters from our data would be a good starting point to study the relationship between STA shape differences and the the range of passive biophysical attributes, this needs further computational modeling efforts. Since the diversity of feature preference by single neurons in the barrel cortex is not completely understood, a quantification of the functional heterogeneity observed in the linear input filter provided by our results is important for creating biophysically realistic models of cortical circuits and for a better understanding of circuit characteristics.

## MCFA-based variance comparison across attribute sets

To determine which attribute set is most informative about neuronal functional heterogeneity, we compared the amount of neuronal population heterogeneity showcased by each attribute set by comparing the amount of private and shared variance explained by each attribute set. To our knowledge, the multi-attribute set comparison has not been done with physiological attributes before. This method provides a structured pathway to understand the limitations of commonly used electrophysiological features in cluster studies and helps to reach a consensus about the choice of attributes to be used for functional classification. We found that linear input filter explained the highest amount of private variance of all the attribute sets, for both excitatory and inhibitory neuronal populations. This is a clear indication of the usefulness of a linear input filter (STA) as an attribute to explore functional heterogeneity. Contrarily, we found that passive biophysical attributes and action potential attributes explain the most shared variance for excitatory and inhibitory populations, suggesting that these attributes are correlated and contain similar latent structures. This is an important result that can aid in the debate around selecting a feature that is the most informative about neuronal functional heterogeneity. We expect our approach to provide a framework for comparing heterogeneity across other brain regions as well.

## Limitations

There are several limitations to our study. Most important limitation is the fact that the sample size for the shared FN and SH comparison set was 186 cells, which might not be enough to capture all variability across the somatosensory cortex layer 2/3. For the second part of the study, the sample size was 312 neurons. We expect an increased sample size would increase confidence in functional neuronal clusters. It would be insightful to sample from all layers of the barrel cortex and compare the linear input filter across layers, to gain a complete insight into the barrel cortical functional diversity. It has been argued that the activity of a large population of neurons that captures a certain behavior can be approximated by a low-dimensional representation from a few neurons [65] and therefore that the number of recorded neurons should depend on the neural task complexity. We also acknowledge that even though the Frozen Noise is a better approximation of the synaptic input than a Step-and-Hold stimulus, it might not be representative of the full dimensionality of the input that a neuron receives in vivo, as the FN input is based on somatic current injection. We understand that a somatic current injection doesn't represent the full range of non-linear dendritic integration but it is quite helpful in estimating the linear input filter of a neurons, which is found to be the most varying across neural populations and is sufficient for understanding the functional heterogeneity. The FN input, although representative of somatic input, lacks the gradual rise and decay dynamic of the input in-vivo and should be addressed in future experimental recording. The E/I labels in this study are based on firing rates and broad/narrow distinction of the action potential waveform, which is a standard practice in the field but can cause ambiguity in classification results, for example there are interneuron types ($SST^+$ specifically) that do not clearly fall into the narrow width types. Therefore, we acknowledge the lack of morphological and transcriptomic labels of the neurons, which makes the clusters found in this study incomparable to the commonly known types [66]. We expect a dynamic clamp-like setup using the frozen noise input along with morphological and transcriptomic labels to provide more clarity. Our study is also limited in providing a mechanistic relationship between the attribute sets we use, such as how the input changes the passive biophysical attributes and ultimately the linear input filter which leads to different clusters. Although the point GLIF model is quite helpful for extracting passive features from recordings with FN like input, it does not provide a mechanistic description at ion channel level resolution for the variability in adaptation current and the feature selectivity. Moreover, passive features such as resistance and capacitance can be more reliably extracted from Step responses. As explained above, we expect that a more detailed single neuron model, when studied with a physiologically realistic input, would provide a more elaborate picture of how the passive biophysical properties give rise to action potential and adaptation properties, which eventually result in the linear input filter. However, more detailed neuron models are difficult to fit and often do not produce unambiguous model properties [67,68].

## Conclusion

In conclusion, we show that neuronal functional classification is a function of the input protocol and therefore, a physiologically realistic input should be preferred for functional classification. We also established that linear input filters are the most distinguishing property, compared to action potential, passive biophysical, and adaptation current attributes, for understanding the functional diversity of neurons when stimulated with a physiologically realistic input. These results provide an important recommendation for neural taxonomists and electrophysiologists: to consider a neuron's physiological input when defining neural identity. We expect computational single neuron as well as cortical network modeling efforts to discover the implications of the heterogeneity found in the 4 attribute sets of the excitatory and inhibitory neuronal populations we studied.

## Supporting information

**S1 Fig. Waveform and Action Potential properties manifold comparison.** (**A**) Overlaid UMAP representation FN and SH waveforms from 186 neuron used in classification. The waveform shapes are different between SH and FN protocols. (**B**) FN and SH Action potential parameters used for classification projected together on the same space. The Action potential properties are different between FN and SH protocols. The SH Action potential properties show a bigger spread than FN properties. (**C**) UMAP projection of averaged Waveform comparison between the first half and the entire trial. The average waveform shape doesn't change as a result of trial length.
(TIFF)

**S2 Fig. Louvain vs Ensemble Clustering for Graphs (ECG) algorithm comparison.** (**A**) UMAP embedding of FN waveforms colored with cluster labels found using Ensemble clustering, with original waveforms in the same color as the respective color. 7 clusters were observed, the same number as in the case of the Louvain community method. (**B**) UMAP embedding of SH waveforms colored with cluster labels found using Ensemble clustering, with original waveforms in the same color as the respective color. 7 clusters were observed, the same number as in the case of the Louvain community method. (**C**) Heatmap showing the correspondence between Louvain and Ensemble clustering for graph on FN waveforms. Clusters using the Louvain community detection algorithm show a high correspondence with the clusters obtained using Ensemble clustering for the graph method. (**D**) Heatmap showing the correspondence between Louvain and Ensemble clustering for graph on SH waveforms. Clusters using the Louvain community detection algorithm show a high correspondence with the clusters obtained using Ensemble clustering for the graph method.
(TIFF)

**S3 Fig. Cluster stability after leaving one attribute out at a time.** (**A-B**) Stability of clusters for action potential attributes leaving one attribute out for excitatory and Inhibitory sets. The clustering was performed 25 times with random 80% samples for resolution attributes ranging from 0.0 to 1, the mean and standard deviation of the resulting number of clusters and modularity score are plotted. For the chosen resolution parameter (1.0), the cluster numbers fluctuate between 5-7. The cluster number fluctuates more for the excitatory set (left) than the Inhibitory set (right) (**C-D**) Stability of clusters for biophysical attributes leaving one attribute out for excitatory and Inhibitory sets. The clustering was performed 25 times with random 90% samples for resolution parameters ranging from 0.0 to 1, the mean and standard deviation of the resulting number of clusters and modularity score are plotted. For the chosen resolution parameter (1.0), the excitatory population fluctuates between 6-8. On the contrary, the Inhibitory population doesn't show any fluctuation.
(TIFF)

**S4 Fig. Diversity of firing rate and AP half-width.** Firing rate vs AP half-width between excitatory (red) and inhibitory (blue) populations. The excitatory population has a lower firing rate and higher AP width. The Inhibitory population has a lower AP width and higher firing rate.
(TIFF)

**S5 Fig. STA Heterogeneity.** (**A**) The scatter plot shows the baseline added to the theoretical input vs the Peak distance of the STA. The excitatory population has a higher baseline and higher peak distance. The variance for both baseline and peak distance seems higher for the excitatory population than inhibitory population. (**B**) The STA cosine similarity is for the excitatory population (top left), the STA cosine similarity is for the inhibitory population (top right), and the STA cosine similarity is between the excitatory and inhibitory populations. The excitatory population has higher similarity than the inhibitory population.
(TIFF)

**S6 Fig. Cluster assignment likelihood across attributes.** (**A-E**) Heatmap showing the likelihood for neuron clustering in attribute 1 clustering in one of the clusters in attribute 2 for excitatory (left) and inhibitory (right) population. For each attribute (Action potential, Passive biophysical, STA, Adaptation current ($\eta$), none of the clusters show a high likelihood to be clustering in another attribute. (**F**) Cluster label comparison between modularity pairs using Mutual information score for excitatory and inhibitory population. None of the pairs reach the modularity of 0.25 showing that neurons cluster differently across attribute sets.)
(TIFF)

**S7 Fig. Trace visualization for FN waveform clusters.** The FN and SH traces for each waveform based cluster is shown on the left and right respectively. The traces are colored based on the color of the FN waveform clusters in Fig 1. The corresponding voltage recording for 400 mA input step is plotted for the SH trial.
(TIFF)

## Author contributions

**Conceptualization:** Nishant Joshi, Tansu Celikel.

**Data curation:** Nishant Joshi.

**Formal analysis:** Nishant Joshi.

**Funding acquisition:** Tansu Celikel, Fleur Zeldenrust.

**Investigation:** Nishant Joshi.

**Methodology:** Nishant Joshi.

**Project administration:** Sven van Der Burg, Tansu Celikel, Fleur Zeldenrust.

**Resources:** Tansu Celikel.

**Software:** Nishant Joshi.

**Supervision:** Sven van Der Burg, Tansu Celikel, Fleur Zeldenrust.

**Validation:** Sven van Der Burg, Tansu Celikel, Fleur Zeldenrust.

**Visualization:** Nishant Joshi.

**Writing – original draft:** Nishant Joshi.

**Writing – review & editing:** Sven van Der Burg, Tansu Celikel, Fleur Zeldenrust.

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
