## [Decision Letter · Decision Letter 0]

18 Sep 2025

PCOMPBIOL-D-25-00863

Neuronal Identity is Not Static: An Input-Driven Perspective

PLOS Computational Biology

Dear Dr. Joshi,

Thank you for submitting your manuscript to PLOS Computational Biology. After careful consideration, we feel that it has merit but does not fully meet PLOS Computational Biology's publication criteria as it currently stands. Therefore, we invite you to submit a revised version of the manuscript that addresses the points raised during the review process.

Please submit your revised manuscript within 60 days Nov 18 2025 11:59PM. If you will need more time than this to complete your revisions, please reply to this message or contact the journal office at ploscompbiol@plos.org. Please include the following items when submitting your revised manuscript:

We look forward to receiving your revised manuscript.

Kind regards,

Elishai Ezra Tsur

Academic Editor

PLOS Computational Biology

Daniele Marinazzo

Section Editor

PLOS Computational Biology

**Journal Requirements:**

2) The resolution of Figure S1-S6 is very low and somewhat difficult to read. It is important that our Editors and Peer Reviewers are able to read all parts of a submission. Please replace this figure with a higher resolution copy.

3) We note that your Data Availability Statement is currently as follows: "All the data and code for this work is completely open source and the links are provided in the manuscript.". Please confirm at this time whether or not your submission contains all raw data required to replicate the results of your study. Authors must share the “minimal data set” for their submission. PLOS defines the minimal data set to consist of the data required to replicate all study findings reported in the article, as well as related metadata and methods (https://journals.plos.org/plosone/s/data-availability#loc-minimal-data-set-definition).

4) Please amend your detailed Financial Disclosure statement. This is published with the article. It must therefore be completed in full sentences and contain the exact wording you wish to be published.

5) Kindly revise your competing statement to align with the journal's style guidelines: 'The authors declare that there are no competing interests.'

**Reviewers' comments:**

Reviewer's Responses to Questions

**Comments to the Authors:**

Reviewer #1: Review of the manuscript “Neuronal Identity is Not Static: An Input-Driven Perspective”

Overall assessment:

The manuscript “Neuronal Identity is Not Static: An Input-Driven Perspective” by Joshi et al. examines how functional classification of rodent neurons depends on the type of stimulation used to probe the neurons’ intrinsic cellular properties and excitability. This effort is well justified for several reasons. In today’s neuroscience, high-throughput techniques of transcriptomics, proteomics and morphological analysis have been steadily gaining dominance over electrophysiological analysis that is aimed at characterizing the neurons’ dynamics and functional properties. At the same time, classification of neuronal populations based on molecular vs. functional properties often result in divergent interpretation of the experimental observations. Additionally, electrophysiological characterization of single neuron function and intrinsic membrane properties is still largely based on conventional static protocols of current clamp. Neurons have been known to exhibit intricate integrative and resonant properties not readily observed under standard current step stimuli. The authors of this manuscript address these questions by comparing results of classification of mouse barrel cortex neurons either stimulated by standard stepwise current commands or by exposing them to physiologically more realistic, noise-like current waveforms. The neuron populations are classified into functional phenotypes and the clustering depends on the type of input the neurons receive. This is an important observation and motivates the use of dynamic, spectrally broadband stimulation in electrophysiological experiments.

I find the results of the statistical analysis convincing, thorough and well documented. The authors employ a wide range of statistical and classification methods and perform additional tests to strengthen their claims. At the same time, the manuscript would benefit from including more data on the types of neurons studied and discussing the relationship between the classification schemes and the intrinsic active and passive membrane properties (e.g. the role of potent voltage-gated currents, like Ih, D-type K-currents, etc.). In fact, the description of mathematical tools and numeric data (including figure legends) is a little lengthy, somewhat redundant, and might be condensed, e.g. sections 0.1.1 and 0.1.2 could be combined. I would also welcome the presentation of additional electrophysiological data, e.g. families of voltage traces recorded from different types of barrel cortex neurons. Such traces can be informative in detecting peculiar features of the membrane potential that are associated with specific voltage-dependent currents.

Specific comments and suggestions:

• Figure legends are extensive and should be condensed to aid readability of the manuscript.

• According to the Step and Hold protocol the neurons were stimulated 10-times, using 40 to 400 pA levels. Did the authors use fixed, depolarizing current steps (no incrementing)? Static current protocols, while certainly not physiologically realistic inputs, can still uncover important biophysical parameters of the stimulated cells, especially when many current levels are applied (e.g. from -200 up to +200 pA). Biophysical features such as membrane resistance, - time constant, voltage sag, inward or outward rectification index, etc. can be readily extracted from the voltage traces recorded in such experiments. The extraction of biophysical features using a GLIF model (line 176) might provide less reliable estimates for the passive membrane properties than the analysis of current step data.

• The analysis of the electrophysiological data focuses on action potential features and temporal parameters of firing (e.g. ISI-associated parameters). Clearly, this is justified, as the process of action potential generation in response to excitatory inputs represent the main function of neurons. At the same time, AP features and intrinsic excitability depend on a multitude of subthreshold voltage-gated currents, too. The impact of such currents are also better estimated from experiments when multiple levels of injected current are used.

• Fig. 1d: It would be useful to show the membrane potential change of the stimulated neuron during the current step including the transients, i.e. the onset and termination phases. Fig. 1c: The top trace is a little weird, very sharp changes in Vm are seen, this can indicate some issues with electrode access resistance or capacitance. The two traces (red and blue) show very different neuronal behavior, but the underlying inputs seem to be different, too. Were the waveforms of the 360-s frozen noise stimuli identical for the different neurons?

• Observed temporal parameters of action potentials such as their amplitude and slope depend on the electrode access resistance and filter parameters used during the recordings. Did the authors check the quality of the recordings and accepted only the ones that meet the criteria for accurate feature exraction?

• Line 154: Interspike intervals of a neuron are primarily influenced by the inputs the neuron receives, so I consider ISIs as kind of vague measures of the neuron’s intrinsic properties. ISIs are strongly determined by the prior history of membrane potential and the instantaneous levels of excitation and inhibition (the same is true for the spike threshold). It is certainly not easy to decide whether parameters of APs can be (should be) averaged and measures of descriptive statistic can be effectively applied to quantify the complex process of neuronal input-output transformation. One can claim that averaging AP parameters results in loss of information that might impact the classification. It might be a good idea to treat each action potential as a separate entity and construct higher dimensional feature vectors from their parameters like voltage threshold, amplitude, half-width, etc.

Reviewer #2: In this paper, the authors perform an extensive analysis aimed at characterizing neuronal identity by using two different types of stimulus: the classical constant injection of current (termed here step-and-hold, SH) and a more biologically realistic stimulus termed frozen noise. In the latter, a current mimicking the simultaneous arrival of a barrage of presynaptic spikes is injected into the cell. The main result of the paper is that neuronal identity strongly depends on the type of stimulus used and should therefore not be treated as an “intrinsic” property of the cell, but rather a continuously changing attribute of each neuron.

The paper is well-written and clear, but while the results and the analyses performed are interesting, there are some points that I think should be addressed before the manuscript reaches a form suitable for publication in PLOS Computational Biology.

The main issue I have with the results as they are currently presented concerns the distinction between (putative) excitatory and inhibitory neurons. While the criterion used by the authors (broad spikes and low firing rates for excitatory neurons vs. narrow spikes and high firing rates for inhibitory neurons) is the classically accepted one, I am afraid that this classification, without any type of information about the genetic identity of the cells under analysis, is too prone to mistakes in attribution to be of practical use. For instance, there are several interneuron classes (e.g., somatostatin-positive neurons like Martinotti cells) that don’t fall neatly in either class (see for instance clusters 2 and 8 in Fig. 3b,c). Therefore, without genetic access to these cells, there is no way of knowing for sure whether such neurons are in the dataset considered by the authors and, if that is the case, in which cluster they have been placed. In my opinion, this greatly reduces the usefulness of the results presented in Figs. 4-7, since one is never 100% that what the authors call “excitatory” or “inhibitory” cells indeed belong to that specific class. Also, I think the authors should justify better why they used features extracted from the FN traces to distinguish between putative excitatory and inhibitory cells, rather than from the SH traces: this is because firing rates in the range 5 to 10 spikes/s can be hardly considered “high frequencies” when parvalbumin-positive basket cells can fire up to 100 spikes/s in response to a DC step of current.

Other concerns I have are the following:

1. The description of the frozen noise protocol is not clear.

2. The authors claim that the FN protocol is more biologically realistic than the step and hold one. While I agree with this claim, the voltage traces shown in Fig. 1c, particularly in the top panel, do not look very realistic either: the jumps in the membrane potential induced by the stimulation appear to be excessively large, especially compared to what one would expect to see in vivo.

3. Since the cluster structure in Fig. 1a-b looks somewhat similar in the FN and SH conditions, I think it would be more clear for the reader if the clusters were numbered and colored in a way that makes interpretation more straightforward: for example, clusters 2 and 6 in the FN condition should have the same numbers and colors as clusters 3 and 4 in the SH condition, respectively. While I realize that this is not possible for all clusters (e.g., cluster 5 in the SH condition does not seem to have a clear equivalent in the FN condition), I think the aim of this reordering should be to maximize the values of cosine similarity on the diagonal of the heat map in Fig. 1f.

Minor points:

1. Define the acronyms only once.

2. The panels in Fig. 1 are not described in order in the text (e.g., panels c-d are mentioned before panels a-b

3. Line 405: “compare” is repeated twice.

4. Line 361 “dimension d”, d should be in italics.

**Have the authors made all data and (if applicable) computational code underlying the findings in their manuscript fully available?**

Reviewer #1: Yes

Reviewer #2: Yes

PLOS authors have the option to publish the peer review history of their article (what does this mean?). If published, this will include your full peer review and any attached files.

Reviewer #1: No

Reviewer #2: **Yes: **Daniele Linaro

**Figure resubmission:**
---

## [Decision Letter · Decision Letter 1]

7 Dec 2025

Dear Mr. Joshi,

We are pleased to inform you that your manuscript 'Neuronal Identity is Not Static: An Input-Driven Perspective' has been provisionally accepted for publication in PLOS Computational Biology.

Best regards,

Elishai Ezra Tsur

Academic Editor

PLOS Computational Biology

Daniele Marinazzo

Section Editor

PLOS Computational Biology

Reviewer's Responses to Questions

**Comments to the Authors:**

Reviewer #1: The Authors responded to my criticism and specific questions, and I believe, their work is suitable in its current form to be published in PLoS Computational Biology. I fully agree with the notion and justification of introducing new concepts and methods in the functional classification of neurons. The manuscript delivers an important point, i.e. functional properties of neurons should be investigated in the context of temporally complex inputs rather than relying solely on conventional static stimulus protocols. This way we can better appreciate the flexible and input-dependent organization of neuronal assemblies.

While the addition of additional data and figures improved the manuscript and I accept the modifications, I suggest the checking of the recordings shown in Supplementary Figure 7. Here, I find action potential peak levels reaching over +75 mV and even over +100 mV in one example. This might be due to incorrect calibration or wrong bridge balance settings. The action potential shape parameters (kinetics, ISIs) are not affected by this technical issue, but passive membrane properties could be calculated incorrectly.

Reviewer #2: The authors have satisfactorily addressed all my concerns and therefore I recommend the paper for publication.

**Have the authors made all data and (if applicable) computational code underlying the findings in their manuscript fully available?**

Reviewer #1: Yes

Reviewer #2: Yes

PLOS authors have the option to publish the peer review history of their article (what does this mean?). If published, this will include your full peer review and any attached files.

Reviewer #1: **Yes: **Attila Szűcs

Reviewer #2: **Yes: **Daniele Linaro

---

## [Editor Report · Acceptance letter]

PCOMPBIOL-D-25-00863R1

Neuronal Identity is Not Static: An Input-Driven Perspective

Dear Dr Joshi,

I am pleased to inform you that your manuscript has been formally accepted for publication in PLOS Computational Biology. Your manuscript is now with our production department and you will be notified of the publication date in due course.

With kind regards,

Aiswarya Satheesan
